# How economically and environmentally viable are multiple dams in the Upper Cauvery basin, India? A hydro-economic analysis using a landscape-based hydrological model

Anjana Ekka[1,3], Yong Jiang [2], Saket Pande[1], Pieter van der Zaag [1,2]

[1] Department of Water Management, Delft University of Technology, Delft, The Netherlands

[2] IHE Delft Institute for Water Education, Delft, The Netherlands

[3] ICAR-Central Inland Fisheries Research Institute, Barrackpore, India

*Correspondence to*: Saket Pande (s.pande@tudelft.nl)

**Abstract.** The construction of dams threatens the health of watershed ecosystems. The purpose of the study is to show how multiple dams in a basin can impact hydrological flow regimes and subsequently aquatic ecosystems that depend on river flows. The approach assesses the ecosystem services, including the tradeoffs between economic and ecological services, due to altered flow regimes. It uses a previously developed model that integrates a landscape-based hydrological model with a reservoir operations model at basin scale. The approach is novel not only because it offers the analysis of alterations in ecosystem services at daily scale when pre-dam data in unavailable but also because dams can be synthetically placed anywhere in the river network and the corresponding alterations in flow regimes simulated in a flexible manner. As a proof of concept, we analyse the economic and ecological performances of different spatial configuration of existing reservoirs, instead of synthetically placed reservoirs, in the Upper Cauvery River basin in India. Such a study is timely and being conducted for the first time, especially in light of calls to assess cascade of reservoirs in India and regions elsewhere where pre-dam data is unavailable. The hydrological impact of different configurations of reservoirs is quantified using Indicators of Hydrologic Alteration (IHA). Additionally, the production of two major ecosystem services that depend on the flow regime of the river, as indicated by irrigated agricultural production and a normalized fish diversity index, is estimated, and a trade-off curve, *i.e.* a production possibility frontier, for the two services is established. Through the lens of the indices chosen for the ecosystem services, the results show that smaller reservoirs on lower-order streams are better for the basin economy and the environment than larger reservoirs. Cultivating irrigated crops of higher value can maximize the value of stored water and, with lower storage, generate similar economic value than with lower value crops while reducing hydrological alterations. The proposed approach, especially when simulating synthetic spatial configurations of reservoirs, can help water and river basin managers to understand the provision of ecosystem services in hydrologically altered basins, optimize dam operations, or even prioritize dam removals with a view to achieve a balanced provision of ecosystem services.

## 1. Introduction

Population growth, economic development, and climate change have necessitated the construction of water storage projects such as dams and reservoirs to meet the societal needs for water, food, and energy, among others (Suwal et al., 2020; Vanham et al., 2011). A large number of cascade reservoirs, i.e. multiple dams constructed along a river network, have already been built and many more are in the process of construction (Suwal et al., 2020). The establishment of such reservoirs and dams alter basin hydrological conditions, particularly river flows downstream of the dams, by storing and releasing river water that can affect aquatic ecosystems in the basin.

Understanding the impact of multiple dams is important for the sustainable development of river basins. The flow regime of rivers is considered a key factor that is affected by dams while determining river ecosystem health (Richter et al., 1996; Brauman et al., 2007). Many scholars have used the degree of hydrological alteration to measure the hydrological impact of dams on aquatic ecosystems (Gierszewski et al., 2020; Lu et al., 2018, Mittal et al., 2016; Song et al., 2020). While hydrological alterations by dams have basin-wide implications, impact assessment typically

concentrates on river segments, assessing the impact upstream or downstream of single dam projects (Nilsson and Berggren, 2000). The assessment becomes more challenging when critical ecosystems are affected by multiple dams, or a cascade of dams, here referred to as a configuration of dams (Arias et al., 2014; Berga et al., 2006).

A viable configuration of dams considers factors such as stakeholder preferences and ecosystem preservation to ensure

a sustainable functioning of a dam system. From a stakeholder perspective, it takes into account the preferences and needs of different parties involved, including local communities, government bodies, environmental organizations, and industries. The aim is to strike a balance among diverse interests, incorporating stakeholder preferences into the design and operation of a dam system (Kemmler & Spreng, 2007). From a phenomenological perspective, a viable configuration respects the boundaries within the ecosystem that, if exceeded, could disrupt the functioning of key

components such as fish biodiversity, aquatic habitats, and downstream water quality (Kumar and Katoch, 2014). Overall, achieving a sustainable balance between societal needs and environmental protection requires careful planning, scientific analysis, and transparent decision-making processes in dam development (Kemmler & Spreng, 2007; Kumar and Katoch, 2014).

There are ecological-economic models that analyse tradeoffs between economic development and ecological conservation or among ecosystem services, but they usually consider the effect of a single reservoir (Lu et al., 2018; Rodríguez et al., 2006; Fanaian et al., 2015) or quantify tradeoffs between energy production and environmental degradation (Null, et al., 2020; Song et al., 2019; Wild et al., 2019; Schmitt et al., 2018). Several studies have targeted multiple dams (Consoli et al., 2022; Van Cappellen and Maavara, 2016; Ouyang et al., 2011; Wang et al., 2019; Zhang

et al., 2020; He et al., 2022). For example, Ouyang et al (2011) studied the impact of cascade dams on streamflow, sand concentration, and nutrient pollutant discharge in the upper reaches of the Yellow river. Similarly, Zhang et al. (2020) focused on understanding the hydrological impact of cascade dams in a small headwater watershed under

climate variability. However, there are no studies that assess the impact of multiple dams on the provision of ecosystem services at macro basin scales and at daily time step when pre-dams data is unavailable. This paper aims to fill this gap by proposing a flexible approach that can simulate the effect of multiple dams on ecosystems services and assess tradeoffs between different ecosystem services competing over river flow under different spatial configurations of dams.

In this study, we have chosen economic value of agriculture production and normalized fish diversity index based on an empirical equation of fish species richness as the indicators of ecosystem services to represent economic development and environmental sustainability respectively. The study area is the Upper Cauvery River basin in India where these ecosystem services dominate. The paper aims to assess how different configurations of existing reservoirs of varying sizes in the basin perform in terms of these ecosystem services so that desirable configurations of reservoirs could be identified. Here a desirable configuration of existing reservoirs is one that efficiently meets agricultural water demand while considering ecological sustainability better than other configurations.

The novelty of the approach is the tradeoff analysis based on model simulations that can simulate not just the effects of various configurations of existing reservoirs at daily scale when pre-intervention data is unavailable but also the effects of synthetic configurations of reservoirs, though the current study focuses only on existing reservoirs as a proof of concept. The approach is based on Ekka et al. (2022) who presented a landscape-based hydrological model coupled with a model of reservoir operations at a daily scale, to primarily analyse the hydrological effects of single reservoirs. In the present study, the existing reservoirs of the Upper Cauvery River basin are integrated to examine their overall effects on dominant ecosystem services at the basin level. For the first time such an assessment of flow alterations due to a cascade of multiple reservoirs is being conducted at a daily time scale for a major river basin in India where pre-intervention data were not available. We will show that this approach can measure the impact of cascade dams on the provision of ecosystem services in basins at a fine temporal resolution and can analyze and optimize dam development that balances the provision of multiple ecosystem services.

The paper is structured as follows. The methodology is discussed in section 2 which includes the integration of reservoirs and the analysis of tradeoff between fish species richness and agricultural production. The results are subsequently presented in section 3 and discussed in section 4. In section 5, the paper concludes with possible future implications of the study for sustainable reservoir management incorporating ecosystem services-based assessments that balance environmental with socio-economic needs.

## 2. Methodology

The aim of the paper is to assess the hydrological, ecological, and economic consequences of multiple dams within the study area. To achieve this objective, a landscape based hydrological model (FLEX-Topo) was integrated with a reservoir operations model. The setup of this model was explained in detail, including its inputs, parameters calibrated and calibration results, in Ekka et al. (2022). This integration involves modeling the operations of the reservoirs, as well as the hydrology of the upstream and downstream areas of the reservoirs (Figure 1). By integrating these models, the impact of reservoirs on the flow regimes downstream and the delivery of ecosystem services can be evaluated (see Figure 4). A detailed description is given below.

### 2.1 Description of the study area

The Cauvery River is the fourth largest river in peninsular India and originates from Talakaveri in the Kodagu district of Karnataka state of India. The river has a drainage area of 81,155 km$^2$, which is nearly 2.7 % of the total geographical area of the country (India WRIS, 2015). The Cauvery basin extends over the Indian states of Karnataka (42 %), Kerala (4 %), and Tamil Nadu (54 %) including the Karaikal region of Puducherry before draining into the Bay of Bengal. The states of Karnataka, Tamil Nadu, and Kerala, along with the union territory of Puducherry, all claim a share of water from the Cauvery River (see supplementary materials, Figure S.1).

Agricultural land is dominant in the basin, with an area of 53,700 km$^2$ (or 66 %), which is followed by a forest area of 16,600 km$^2$ (or 21 %) (Sreelash et al., 2014). Along certain stretches of the Cauvery River, extensive abstraction of water is carried out for intensive agriculture (Vedula, 1985; Bhave et al. 2018). Paddy is the most significant irrigated crop in this region, while Ragi, Jawar, and other millets are important rainfed crops. More than 60 % of the total population in the Cauvery basin lives in rural areas with crop-based agriculture as the main occupation (Singh, 2013).

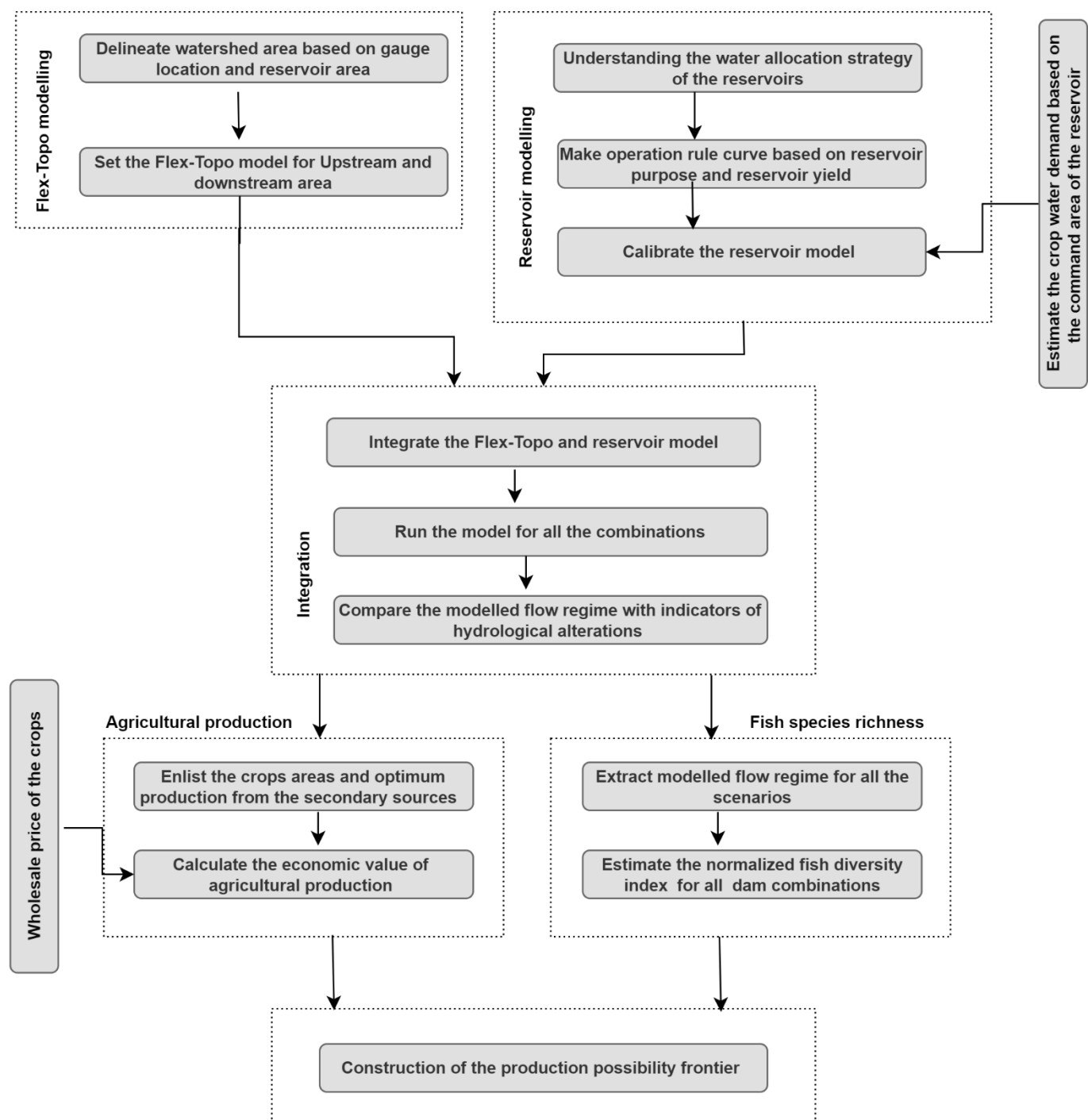

 Figure 1. Overview of the methodological structure of the study

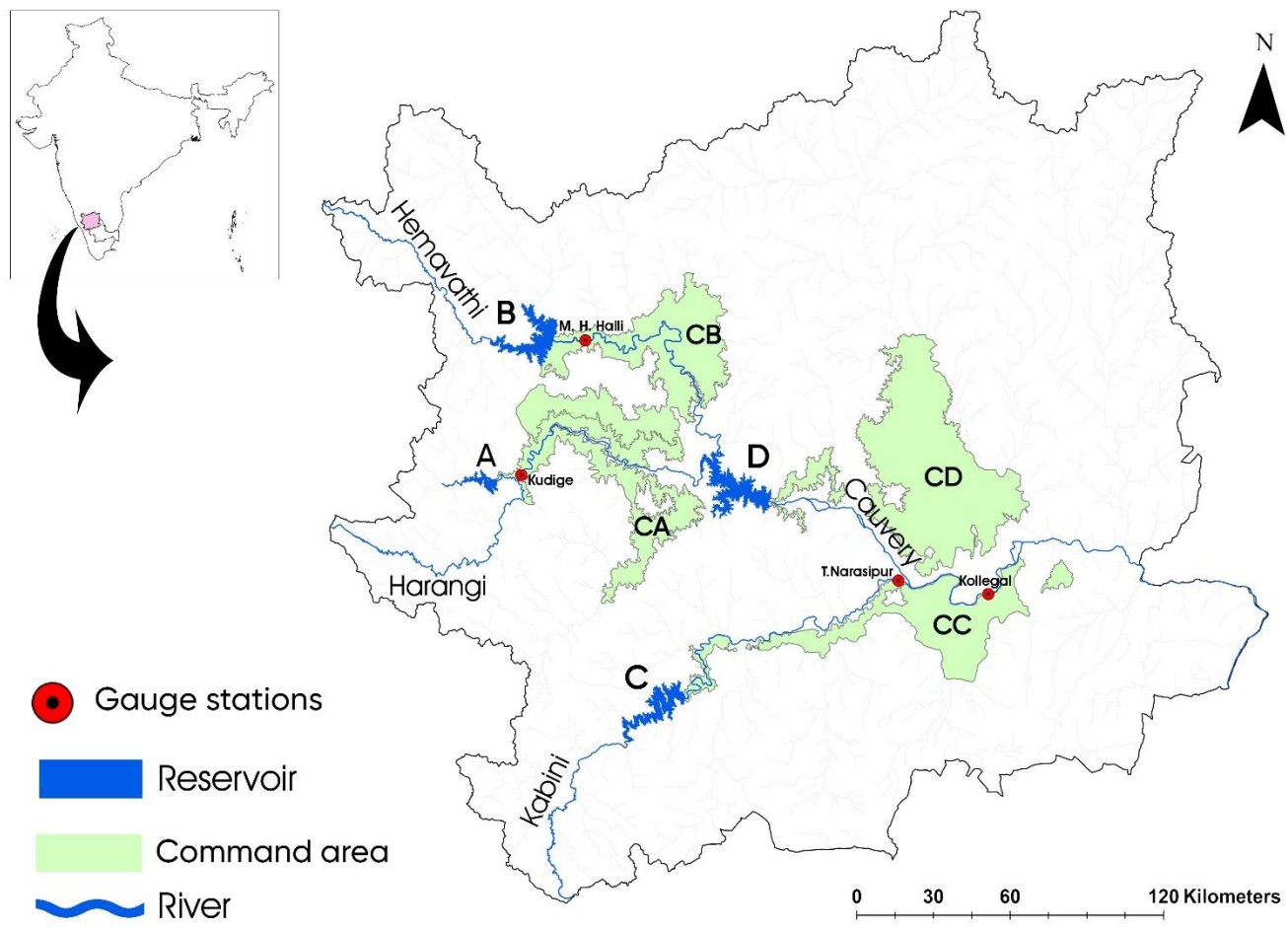

Figure 2. An overview of the Upper Cauvery River Basin. The reservoirs in the study area are labelled as A, B, C, and D, representing Harangi, Hemavathi, Kabini, and Krishna Raja Sagara (KRS) reservoirs respectively. The labels CA, CB, CC, and CD are used to denote the respective command areas[1] served by these reservoirs.

Based on the availability of the data needed for the study and the location of large reservoirs in the basin, the four largest reservoirs in the Upper Cauvery by gross storage capacity are selected for investigation, including Harangi, Hemavathi, Kabini, and Krishna Raja Sagara (KRS) (Figure 2). Among the selected reservoirs, Harangi is the smallest reservoir and KRS is the largest reservoir in terms of gross storage capacity and contributing catchment area (Figure 3).

---

[1] A command area is the area which can be physically irrigated from a reservoir and is fit for cultivation.

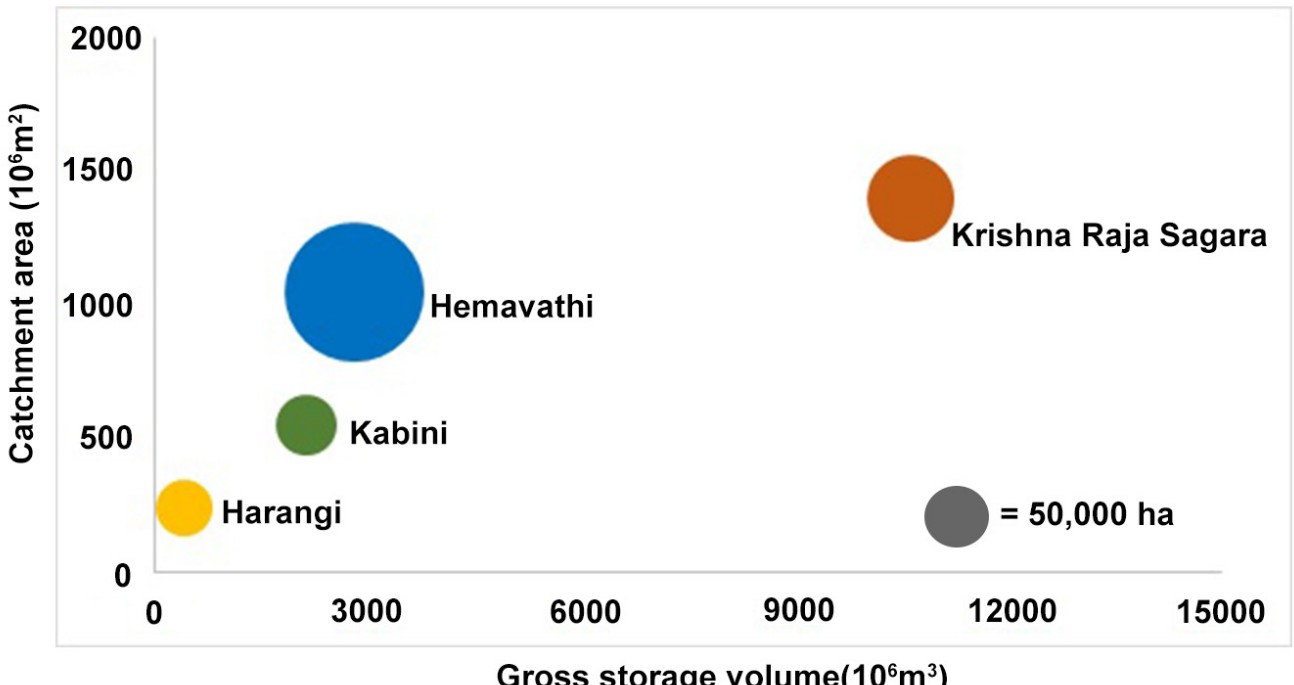


Figure 3. Overview of selected reservoirs by contributing catchment area and gross storage volume. The size of the bubbles is proportional to the size of the command areas (areas under irrigation from water from the reservoirs). The size of the grey bubble is equivalent to 50,000 ha.

**2.2  Hydrological model (The FLEX-Topo Model)**

The present study utilizes a hydrological model called FLEX-Topo (see Supplementary materials section 1 and Figure S.2; Gharari et al., 2014). This parsimonious modeling approach has demonstrated its ability to simulate streamflows in data-scarce basins, as its structure is constrained by topography, requiring relatively few calibration parameters, and yielding reliable flow simulations even under changing land-cover conditions (Gao et al., 2014; Savenije, 2010).

The FLEX-Topo model classifies the landscape of a basin into various Hydrological Response Units (HRUs) based on elevation (Digital Elevation Model - DEM), slope, and Height Above Nearest Drainage (HAND), see section 2.2.1, and HRU specific processes are modelled to simulate river flows. FLEX-Topo is then integrated with a reservoir operations model, which simulates altered flows at daily time steps (see Figure 4).

**2.2.1    Creation of Hydrological Response Units (HRU)**

A Hydrological Response Unit or HRU represents a distinct landscape element assumed to exhibit specific hydrological responses and is accordingly modelled by FLEX-Topo. Its characteristics are influenced by both topography and land use. The topographical aspects, such as plateau, hillslope, and wetland, determine the HRU's

streamflow responses to rainfall. Additionally, the land use, whether forests or agriculture, impacts the HRU's surface conditions, water infiltration rates, and evapotranspiration, further shaping its hydrological response.

For the present study, the classification of landscape into HRUs involves utilizing Digital Elevation Model (DEM), slope, and Height Above the Nearest Drainage (HAND), into three distinct classes, namely hillslopes, plateaus and

wetlands. The slope and HAND data are processed using an 80-meter resolution DEM. The delineation of a sub-basin with a reservoir within is determined based on the location of a streamflow gauge downstream of the reservoir. As Figure 4 shows, the area upstream of the reservoir that is contributing flow to it (known as F1) is delineated by the location of the corresponding dam. Subsequently, the area downstream of the dam directly contributing flow to the gauge (known as F2 in Figure 4) is obtained by clipping F1 from the entire sub-basin delineated with respect to the

gauge. The HRUs are identified for both F1 and F2 contributing areas and subsequently used to execute the FLEX-Topo model for the sub-basin.

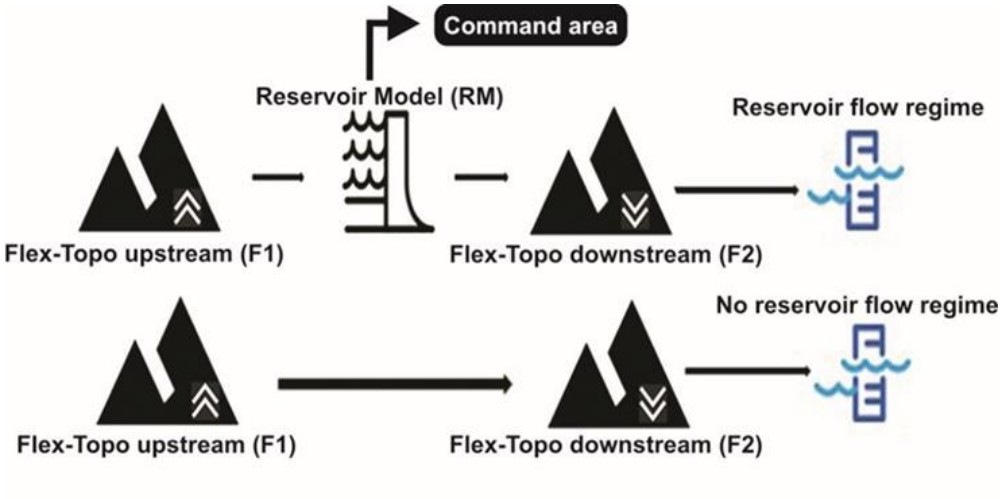

Source: Ekka et al., 2022


Figure 4. Modelling concept for the individual FLEX-Topo-reservoir model: Upstream and downstream areas of the reservoir contributing to a streamflow gauging location downstream of the reservoir (where flow regime is being

observed) are modelled as upstream (F1) and downstream (F2) models respectively. The top panel shows that the reservoir operations model (RM) that contributes to irrigating a certain command area is integrated with F1 and F2

and calibrated. The bottom panel shows how the pre-dam situation is simulated, simply by removing RM from the calibrated model, along with its contribution to irrigate the command area.

### 2.2.2    Forcing data

Rainfall and potential evapotranspiration data are utilized as the forcing data. Daily gridded rainfall data with a spatial

resolution of 0.25° x 0.25° and temperature data with a resolution of 1° x 1° are obtained from the Indian Meteorological Department, Government of India (Pai et al., 2014; Shrivastava et al., 2009). Runoff data is obtained from the Central Water Commission, Government of India. The information on reservoirs, including inflows, outflows, and storage levels, is accessed from the Karnataka State Natural Disaster Monitoring Centre, Government of Karnataka, India, through their official website (https://www.ksndmc.org/Reservoir_Details.aspx). For reservoir

model calibrations, only a time series of six years of daily inflows, storage and outflows was accessible. However, extended periods of streamflow data for the corresponding downstream gauges, rainfall and temperature data for the sub-basins were available. Thus, the six-year reservoir data was used to calibrate the reservoir operations models and the other streamflow and input forcing data were utilized to calibrate the integrated FLEX-Topo and reservoir operations models.


To analyse agricultural production, the data on the cultivated area and average production of crops at the district level in the study area are sourced from the Directorate of Economics and Statistics, Government of Karnataka (https://des.karnataka.gov.in/info-2/Agricultural+Statistics+(AGS)/Reports/en). Additionally, price information for crops in each district is obtained from the website https://agmarknet.gov.in/.


### 2.3 Reservoir operations model

The operation of multi-purpose reservoirs is governed by the objective of meeting the demands of end-users based on certain allocation priorities. The conservation of mass equation (eq. 1) governs each time step:

$$\frac{S_{t+1}-S_t}{\Delta t} = I_t + O_t - E_t + P_t - (L_t * D_t) \qquad (1)$$

where $S_t$ = storage (m$^3$), $I_t$ = Inflow (m$^3$ day$^{-1}$), $O_t$ = outflow (m$^3$ day$^{-1}$), $E_t$ = evaporation on reservoir surface (m$^3$ day$^{-1}$), $P_t$ = precipitation on reservoir surface (m$^3$ day$^{-1}$), $L_t$ = fraction supply of the demand for the reservoir on day t, $D_t$ = demand for river water on day t (m$^3$ day$^{-1}$), and $\Delta t$ = 1 day. The reservoir model is embedded in the FLEX-Topo model by using the modelled outflow from the upstream area as inflow into the reservoir and using the modelled reservoir outflow as inflow to the downstream contributing area in order to model the runoff at a gauge station.

The reservoir operation is based on shortage rule curves that define zones within which specified proportions of the demand are covered (Basson et al., 1994). The reservoir operating rules determine $L_t$. $D_t$ is determined based on water demand calculation for irrigating crops in command areas or for generating hydropower (see Ekka et al., 2022 for further details).

### 2.4 Hydrological-reservoir model simulation (calibration and validation)

The reservoir models were first calibrated using the dataset composed of inflow, outflow, storage, rainfall, and potential evapotranspiration, for the four reservoirs covering the period from January 2011 to December 2016. These were embedded into the FLEX-Topo models of the corresponding sub-basins as mentioned above and the FLEX-Topo parameters were then calibrated. To calibrate the FLEX-Topo parameters, the dataset of rainfall and potential evapotranspiration for the period January 1991 to December 2010 was used. The performance of the integrated model in different sub-basins were then validated using the dataset from 2011 to 2016.

The Elitist Non-Dominated Sorting Genetic (NSGA-II) algorithm was used to calibrate the model parameters (Deb et al., 2000). Two objective functions are defined and minimized simultaneously. The first objective ($f_1$) is the negative of Nash-Sutcliffe Efficiency (-NSE) and the second objective ($f_2$) is the Mean Absolute Error (MAE). Note here that when -NSE is being minimized, NSE is being maximized.

$$f_1 = -NSE = -1 + \frac{\sum_{i=1}^{n}(Q_i^m - Q_i^o)^2}{\sum_{i=1}^{n}(Q_i^o - \bar{Q}_o)^2} \tag{2}$$

$$f_2 = \text{MAE} = \frac{1}{n}\sum_{i=1}^{n}|Q_i^o - Q_i^m| \tag{3}$$

Here, $Q_i^m$ is the $i^{th}$ observation for the observed discharge (mm day$^{-1}$) being evaluated. $Q_i^o$ is the $i^{th}$ value of the modelled discharge (mm day$^{-1}$). $\bar{Q}_o$ is the mean of observed discharge (mm day$^{-1}$) and $n$ being the total number of observations. The mm day$^{-1}$ value of the observed and modelled discharge is obtained by dividing the volumetric flow rate by the area contributing flow to the gauge station. The details of the parameters calibrated for the FLEX-Topo model and the reservoir operation model are provided in Supplementary materials. Also, the NSGA-II parameter setting are detailed in the Supplementary materials; see Tables S.1, S.2 and S.3.

**2.5  Simulating the effects of different spatial configurations of the reservoirs**

Figure 5 shows one specific example of how the effect of various spatial configurations of reservoirs on flow regimes are simulated at the most downstream gauging station. This example considered the spatial configuration that contains all the reservoirs in the basin. The outflows from reservoirs Harangi and Hemavathi flow through the gauge stations of Kudige and M.H. Halli, respectively, and then into the KRS reservoir. Similarly, the outflow from the reservoir Kabini flows through the gauge station T. Narasipur and then joins the outflow from the reservoir KRS at the gauge station Kollegal, which is the most downstream gauging station. The integrated models corresponding to the sub-basins delineated by each of the gauge stations simulate the 'altered' flows reaching at their respective stations.

For example, the sub-basin corresponding to KRS is delineated by the gauging station Kollegal. Hence the flows modelled at this station are considered, including the flows generated by contributing areas corresponding to gauge stations Kudige, M.H. Halli and T. Narasipur where corresponding modelled flows are considered. Such models of flows (with or without respective reservoirs) at the gauge stations downstream of each of the four reservoirs, instead of observed flows, are used for simulating flow regimes at the gauging station Kollegal for various possible configurations of reservoirs upstream.

A total of 16 different configurations were generated by removing one or more reservoirs from the schematic graph presented in Figure 5, and corresponding flows were modelled to simulate flow at the gauge station Kollegal (see Table 1 for an overview of the different configurations). The modelled flows were then compared to understand the

impacts of reservoirs of varying configuration on the flow regime and, subsequently, on the production of the considered ecosystem services that are dominant in the basin (see Table 1).

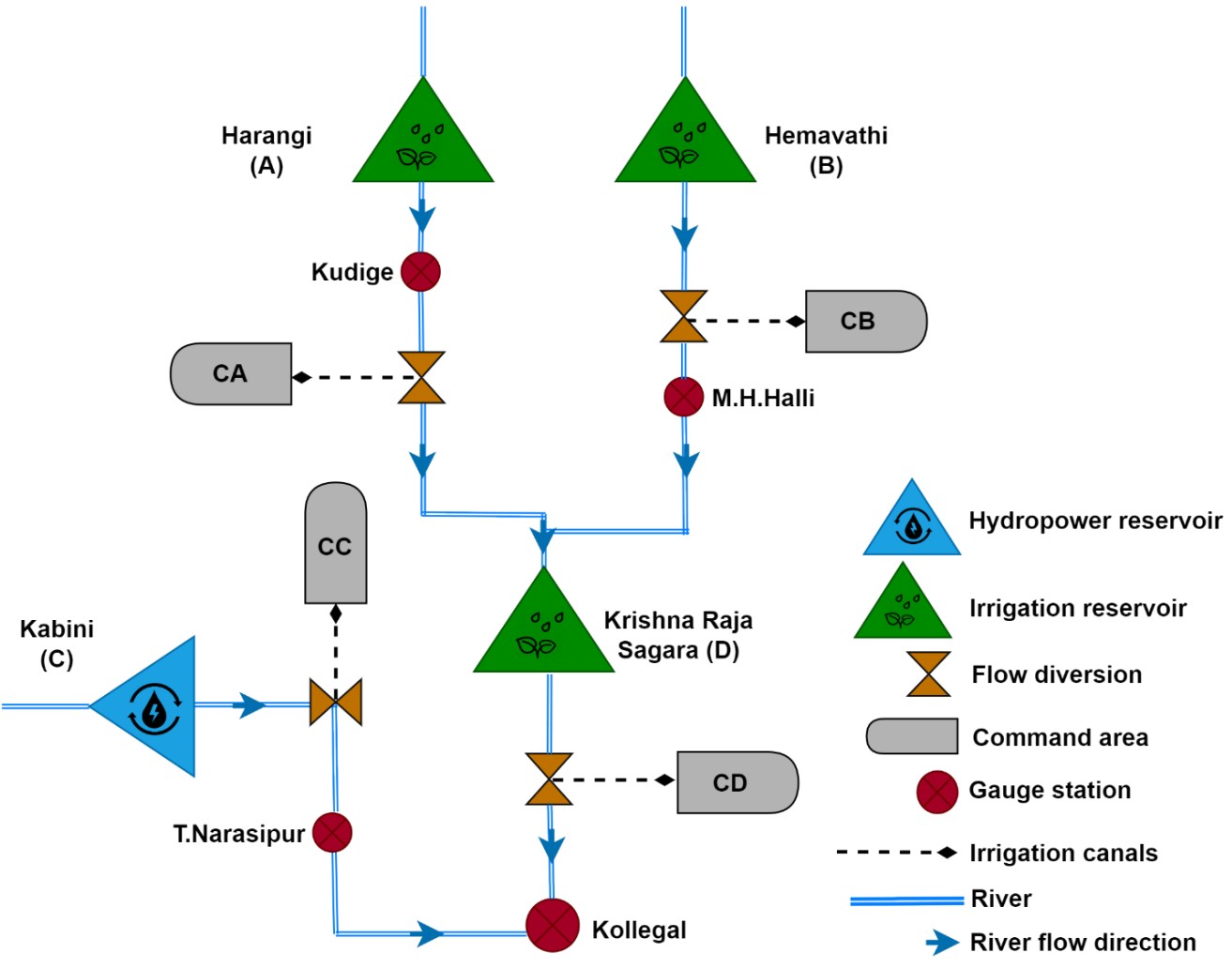


Figure 5. Showing the spatial configuration that contains all four reservoirs of the basin. A reservoir or a combination of reservoirs can be removed from this configuration to simulate correspondingly altered flow regime at Kollegal, the most downstream gauging station location. In this way the reservoirs in different spatial configurations are integrated

together to assess the effect of the configuration on the flows most downstream at Kollegal. All possible configurations of the reservoirs were considered to create a total of 16 different scenarios.


Table 1. Comparison of different configurations of reservoirs by storage volume, purpose, sub-basin area and spatial

configurations.

| Scenarios | Reservoir configurations | Reservoir characteristics | | |
|---|---|---|---|---|
| | | Storage volume ($10^6$ m³) | Purpose of the reservoir & Net Command Area (NCA) | Spatial configuration |
| **Scenario with four reservoirs (Base scenario)** | | | | |
| $S_{abcd}$ | A+B+C+D | A: 240.69<br>B: 1050<br>C: 552.74<br>D: 1400.31 | Irrigation - A, B, D<br>Irrigation & Hydropower-C<br>For individual reservoir<br>NCA : 499,215 ha | A, B: upstream & on a tributary<br>C: downstream & on a tributary<br>D: downstream & on main channel |
| **Scenario with three reservoirs** | | | | |
| $S_{bcd}$ | B+C+D | B: 1050<br>C: 552.74<br>D: 1400.31 | Irrigation - B, D<br>Irrigation & Hydropower-C<br>NCA: 445,677 ha | B: upstream & on a tributary<br>C: downstream & on a tributary<br>D: downstream & on main channel |
| $S_{abd}$ | A+B+D | A: 240.69<br>B: 1050<br>D: 1400.31 | Irrigation - A, B, D<br>NCA: 453,485 ha | A, B: upstream & on a tributary<br>D: downstream & on main channel |
| $S_{acd}$ | A+C+D | A: 240.69<br>C: 552.74<br>D: 1400.31 | Irrigation - A, D<br>Irrigation & Hydropower-C<br>NCA: 207,350 ha | A: upstream & on a tributary<br>C: downstream & on a tributary<br>D: downstream & on main channel |
| $S_{abc}$ | A+B+C | A: 240.69<br>B: 1050<br>C: 552.74 | Irrigation - A, B<br>Irrigation & Hydropower-C<br>NCA: 391,133 ha | A, B: upstream & on a tributary<br>C: downstream & on a tributary |
| **Scenario with two reservoirs** | | | | |
| $S_{bd}$ | B+D | B: 1050<br>D: 1400.31 | Irrigation - B, D<br>NCA:  399,947 ha | B: upstream & on a tributary<br>D: downstream & on main channel |
| $S_{cd}$ | C+D | C: 552.74<br>D: 1400.31 | Irrigation - D<br>Irrigation & Hydropower-C<br>NCA:153,810 ha | C: downstream & on a tributary<br>D: downstream & on main channel |
| $S_{ad}$ | A+D | A: 240.69<br>D: 1400.31 | Irrigation - A, D<br>NCA: 161,620 ha | A: upstream & on a tributary<br>D: downstream & on main channel |
| $S_{cb}$ | C+B | C: 552.74<br>B: 1050 | Irrigation - B<br>Irrigation & Hydropower-C<br>NCA: 153,812 ha | C: downstream & on a tributary<br>B: upstream & on a tributary |
| $S_{ab}$ | A+B | A: 240.69<br>B: 1050 | Irrigation - A, B<br>NCA: 345,403 ha | A, B: upstream & on a tributary |
| $S_{ac}$ | A+C | A: 240.69<br>C: 552.74 | Irrigation - A<br>Irrigation & Hydropower-C<br>NCA: 99,268 ha | A: upstream & on a tributary<br>C: downstream & on a tributary |
| **Scenario with one reservoir** | | | | |
| $S_d$ | D | D: 1400.31 | Irrigation - D<br>NCA: 108,082 ha | D: downstream & on the main channel |
| $S_b$ | B | B: 1050.00 | Irrigation - B<br>NCA: 291,865 ha | B: upstream & on a tributary |
| $S_c$ | C | C: 552.74 | Irrigation & Hydropower-C<br>NCA: 45,730 ha | C: downstream & on a tributary |
| $S_a$ | A | A: 240.69 | Irrigation - A<br>NCA: 53,538 ha | A: upstream & on a tributary |
| **Scenario with no reservoir** | | | | |
| $S_0$ | NO | -- | -- | -- |

## 2.6  Indicators of Hydrological Alteration (IHA)

The set of Indicators of Hydrological Alteration (IHA) initially proposed by Richter et al. (1996) is used to measure the effects of different reservoir configurations on the flow regime in the Upper Cauvery basin. The parameters considered in IHA have strong relationships with river ecosystems, and therefore can be used to assess the impacts of dams on the flow regime. The IHA are classified into five groups based on magnitude of monthly flows, magnitude and duration of annual extreme flow conditions, and frequency and duration of high and low flow rates. Major

indicators used in the study include mean annual discharge, low flows, high flows, low pulse rate, and high pulse rate (detailed definition is provided in Supplementary materials S.4). High frequencies of flows, and alterations of it, can be considered within the IHA given that modeled flow regimes are at daily time scale. Although earlier methods of assessing the impact of impoundments on river channels have involved field surveys, statistical analyses (Yan, 2010), and geomorphic change detection tools (Wheaton, 2015), the IHA framework provides a more systematic assessment

of changes in flows. Its application has been relatively limited in the studies of Indian rivers (Mittal et al., 2014; Kumar and Jayakumar, 2020; Borgohain et al., 2019), often due to lack of pre-dam data availability. The simulations of pre-interventions flows presented here makes this possible, especially when.

### 2.7 Tradeoff between ecosystem services: construction of the Production Possibility Frontier

The production possibility frontier (PPF), also known as the production possibility curve or boundary, is a graphical representation of the different combinations of goods or services that an economy can efficiently produce given its limited resources and technology (Martinez-Harms et al., 2015; King et al., 2015; Cavender-Bares et al., 2015). It can be described as the outward boundary of the convex hull of the production set of the economy. It shows the maximum level of one good or service that can be produced in relation to the production of another good or service, given the

existing resources and technology.

In the Cauvery basin, approximately 48 percent of the land is used for crop cultivation (Singh, 2013). In certain stretches of the Cauvery River, there is extensive water abstraction for intensive agriculture (Vedula, 1985; Bhave et al., 2018). This water extraction has resulted in notable changes in the composition of aquatic species, primarily due

to the construction of reservoirs, and in the overall biodiversity of the river ecosystem. This tradeoff between the corresponding dominant ecosystem services that are provided by the bioeconomy of the basin is represented by a tradeoff between indicators of agricultural production value and fish species richness respectively, and conveniently

represented by the PPF. The value of crop production that dominates the agricultural production value is used for the former, and a specific indicator for fish species richness, namely the normalized fish diversity index, is used for the latter.

Different spatial configurations of the reservoirs correspond to different partitioning of flows for irrigation and for aquatic ecosystems. Therefore, different pairs of crop production values and normalized fish diversity index are generated for different reservoir configurations. Since only existing reservoirs are considered, a production set is determined based on the production outputs of all possible spatial configurations of existing reservoirs. Specifically, it is defined by the convex-hull of the 16 pairs of agricultural production and normalized fish diversity index values, corresponding to the 16 possible spatial configurations of the reservoirs. The production possibility frontier is then the outward boundary of the production set. However, note that this production set can be exhaustively populated by simulating synthetic configurations of artificial reservoirs on the river network. This is left for future work.

### 2.7.1   Agricultural production

The available information on agricultural crops and their distribution is organized at the district level (the lowest administrative level within the boundaries of the states that fall in the basin where such information is available). All the calculations related to these crops are performed at this level, where a total of nine districts are considered in the analysis. The districts falling within each sub-basin of the Upper Cauvery basin are identified and their areas are determined. Subsequently, using the available data, the areas of irrigated and unirrigated land within and outside the sub-basins are calculated (see Supplementary materials, Table S.7). Based on the known cropping patterns for each district, the crops grown are categorized into four growing seasons: kharif (June-September), rabi (October-January), summer (February-May), and perennial  crops. The area dedicated to each crop within a sub-basin is determined proportionally by the acreage of different crops in each district within the sub-basin. The maximum yield under irrigated condition and crop prices are obtained from agricultural census sources. Additionally, information on crop coefficients and crop yield response factors is gathered from published literature (see Supplementary materials, Table S.6). An average yearly real price is estimated for each crop in all the districts within the studied basin (see Supplementary materials, Table S.8). For irrigated areas, the maximum (optimum) yield values from the literature are

used to calculate crop production. However, for unirrigated areas the reduction in yields are estimated based on the actual evapotranspiration estimates of the integrated model.

For agricultural production, the relationship between crop yield and water depends on the corresponding relative reduction in evapotranspiration (ET). The actual yield is calculated based on the following formula by Smith & Steduto

345   (2012)

$$1 - \frac{Y_a}{Y_o} = K_y \left(1 - \frac{\sum_i^n ET_a^i}{\sum_i^n ET_p^i}\right)$$

(4)

where $Y_a$ = actual Yield (kg ha$^{-1}$year$^{-1}$), $Y_o$ = optimum Yield (kg ha$^{-1}$ year$^{-1}$), $ET_a^i$ = Total actual evapotranspiration

for day $i$ out of $n$ days of the crop season (mm day$^{-1}$), $ET_p^i$ = Total potential evapotranspiration for day $i$ out of $n$ days of the crop season (mm day$^{-1}$), and $K_y$ = yield response parameter (-).

The equation 4 presents end-of-season yield as a fraction of optimal yield that depends on how much daily evaporation is accumulated by the crops over the season compared to the respective evaporation demands (optimal evaporation).

Yearly production value is obtained by multiplying the average area of each crop with average simulated yields and prices over 2011-2016. Yields are multiplied by the area cultivated with corresponding crops to calculate the agricultural output; irrigated output if irrigated else rainfed output. Total agricultural production equals the agricultural output from both rainfed and irrigated areas. The crop specific prices are multiplied by the corresponding production output to indicate the economic value of the ecosystem service supported by the basin.


### 2.7.2    Normalized Fish Diversity Index

Aquatic ecosystem health serves as a comprehensive reflection of the physical, chemical, and biological integrity of river ecosystems (Chen et al., 2019; Aazami et al., 2019). Previous studies have investigated various factors to identify the key determinants of river ecological health, including benthic macroinvertebrates, river habitat conditions, and

water quality parameters (Chen et al., 2019). However, when considering biological indicators, fish health becomes crucial as it directly links to the provisioning of services such as food and human health. Fish species richness refers to the number of different fish species present in a particular area or ecosystem. It is one of the indicators of biodiversity and represents the diversity of fish species within a given habitat or geographical region. Species richness is commonly used to assess the ecological health and complexity of aquatic ecosystems (Xu et al.,1999). Therefore,

fish species richness is chosen as the indicator of river health, reflecting the overall health of the aquatic ecosystem. No particular specific fish species is targeted in this study. Fish migration patterns have not been included due to data limitations which includes tracking efficiency, sample bias, limited spatial coverage, as well as species-specific challenges (Planque et al., 2011; Elsdon et al., 2008).

Species-discharge models, based on mean river discharge, are often used to quantify the impact of anthropogenic modification of rivers on species richness (Xenopoulos and Lodge, 2006). However, the flow regime of a river is composed of several ecologically relevant flow characteristics such as magnitude, frequency, duration, timing, and rate of change of flow events that impact species richness. In other words, flow characteristics other than mean river discharge also play a vital role in sustaining aquatic ecosystems. Many Species Discharge Relationship (SDR) models

have been derived based on data of large basins (>500 km$^2$) globally to explain long-run riverine fish species richness (FSR) as a function of discharge and other variables (Schipper and Barbarossa, 2022; Xenopoulos and Lodge, 2006; Iwasaki et al., 2012). In the present study, the basin is >10,000 km$^2$, at which scale discharge is a key variable explaining differences in species richness (Schipper and Barbarossa, 2022, page 1502). We adopted an empirical function (equation 5) by Iwasaki et al. (2012) to quantify fish species richness. We use the equation to assess changes

in fish species due to changes in flow characteristics for the same basin (keeping area and latitude constant to incorporate the fixed effect of the basin). This is very similar to the use of the Budyko curve derived from basin data sets across the globe in hydrology, e.g. space for time substitution to assess the impacts of changes in precipitation on rainfall partitioning in basins in the long run (Bouaziz et al., 2022). Indicators for flow characteristics, such as coefficient of variation of mean frequency of low flow in a year, coefficient of variation in the Julian date of annual

minimum flow, and maximum proportion of the year in which floods have occurred, are also used. Here floods are defined as events when flows are greater than or equal to flows with a 60 % exceedance probability (Olden and Poff, 2003). This choice of a regression equation (equation 5) was suitable for our analysis since the underlying model does not consider water quality and other aspects.

Therefore, only the possible combinations in which the current four reservoirs can appear in the basin are considered as counterfactuals and it is assumed that these dominantly lead to changes in streamflows that in turn influence the variability of FSR based indicators of environmental quality. Further, the equation is not used for predicting FSR but

for an index of environmental health in a two-dimensional tradeoff analysis of dominant ecosystem services that are affected by plausible reservoir scenarios dominantly affecting streamflow.


$$FSR = \exp\ (3.950 - 0.034 * LAT\ +\ 0.273 * Area\ +\ 0.373 * MAD - 1.570 * FL2\ +\ 0.832 * TH3 - 0.116 * TL2)$$

(5)

Where, LAT = Absolute value of the latitude of the gauge station where flow is measured

Area = $\log_{10}$ transformed basin area (km$^2$)

MAD = $\log_{10}$ transformed mean annual discharge (m$^3$s$^{-1}$)

FL2 = Coefficient of variation of mean frequency of low flow per year (-)

TH3 = Maximum proportion of the year (number of days /365) during which floods have occurred (-)

TL2 = Coefficient of variation in the Julian date of the annual minimum flow (-).

The fish species richness index is then normalized into an index, called the "normalized fish diversity index" ($NFDI_i$), for any $i^{th}$ scenario calculated as:

$$NFDI_i = \frac{FSR_i}{\max_i\{FSR_1,..,FSR_i,..,FSR_I)}$$

Where,

$NFDI_i$ is the Normalized Fish Diversity Index for the $i_{th}$ scenario

$FSR_i$ is the Fish Species Richness for the $i_{th}$ scenario

$I = 16$ is the number of scenarios of possible reservoir combinations (counterfactuals)

Utilizing the normalized fish diversity index in our analysis helps reduce dependence on absolute FSR numbers and their changes over different scenarios. Rather than focusing solely on numerical values, our methodology prioritizes

the relative ranking within the tradeoff space. By incorporating proxies for environmental quality and agriculture, this normalization approach facilitates a nuanced assessment. It highlights the relative positions of various scenarios, providing insight into their impacts on both environmental quality and agricultural production.

Due to limitations on the years for which crop prices were available, we used 6 years of simulations 2011-2016 to

estimate flow-related quantities needed in Equation 5. Daily-scale simulations are used for calculating FSR parameters like TH3, FL2, and TL2, along with mean annual flow calculations and evaporation deficit in yield estimations. Daily-scale modeling facilitates space-time substitution in SDR-based FSR, enabling assessment of agricultural production trade-offs with reservoir combinations. In these scenarios, other factors are assumed constant

**3. Results**

This section first reports on the quality of the model developed for the study area. Table S.3 reports on the calibration and validation performance of the model developed for the study area in Ekka et al. (2022). The model was calibrated using the NSGA II multi-objective optimization algorithm, and the Pareto front ranges for both -NSE (Nash Sutcliffe Efficiency) and MAE (Mean Absolute Error) are shown (see supplementary materials for more details). Note the use

of negative NSE was used alongside MAE to minimize the two objective functions jointly (and therefore maximize NSE alongside minimization of MAE). The developed model is then used to simulate flow regimes for the 16 scenarios of different spatial configurations of existing reservoirs as shown in Table 1, and the degree of hydrological alterations is assessed. The production of considered ecosystem services is then quantified, and a production possibility frontier for the considered ecosystem services is derived and discussed.


**3.1 Impact on flow regimes generated by different spatial configurations of reservoirs**

The flow regimes corresponding to different spatial configurations (also referred to as scenarios, see Table 1) of the existing reservoirs are analysed to understand the impact of the latter on the former, utilizing major hydrological

indicators like mean annual flow and annual extreme flow conditions. Additionally, the analysis involves classifying the flow regimes based on the storage volumes of the reservoirs and its uses. All the hydrological indicators are calculated based on the discharges that are simulated at the most downstream (Kollegal) gauge station.

**3.1.1   Flow regimes characterized by storage volumes under different scenarios**

The highest mean annual flow was estimated for $S_0$ (1,548 $m^3s^{-1}$) with no reservoir, followed by $S_c$ (1,460 $m^3s^{-1}$) and $S_b$ (1,377 $m^3s^{-1}$) that are configurations containing only one reservoir (Figure 6). In terms of storage volume, KRS (D) is the biggest reservoir followed by Hemavathi reservoir (B) and Kabini reservoir (C). KRS (D) in the spatial

configurations with one other reservoir ($S_{bd}$, $S_{cd}$, $S_{ad}$) and two other reservoirs ($S_{bcd}$, $S_{abd}$, $S_{acd}$) yielded mean annual flows of less than 500 m$^3$s$^{-1}$. Figure 6 shows the mean monthly variation in the flow for all 16 combinations.

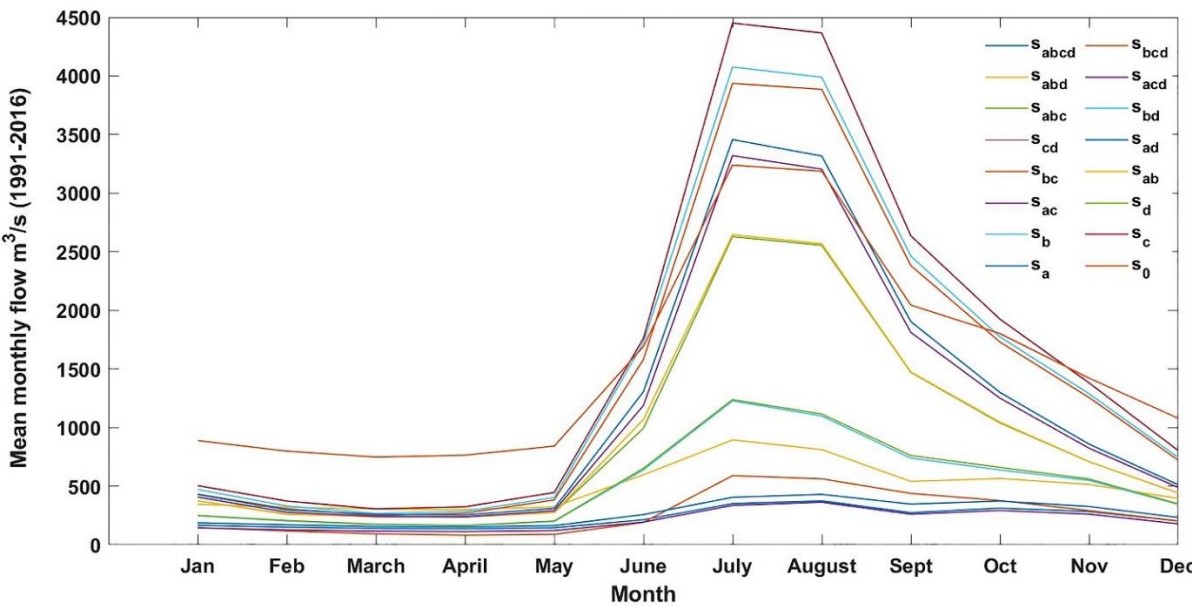


Figure 6. The mean monthly flows resulting from different configurations of reservoirs

Figure 7 shows that the magnitude of annual extreme conditions, the 1-3-7-30-90 day minimum and maximum flows, were greatly affected by the construction of reservoirs having bigger storage volumes. However, in scenarios of

configurations with three reservoirs, $S_{abd}$ has less impact compared to $S_{acd}$ despite Kabini (C) having less storage capacity compared to the Hemavathi reservoir (B).

The extreme low peak flow for scenario $S_D$ appears to be the lowest of the configurations with only one reservoir (Table 2) as KRS (D) reservoir has the largest storage capacity. Similarly, the KRS (D) generated flows with lowest

values of extreme low peak conditions in spatial configurations with two ($S_{bcd}$, $S_{abd}$) and three ($S_{abcd}$) reservoirs. However, in scenarios involving one or two reservoirs despite having varying storage capacities, the extreme low peaks of flows generated by $S_a$, $S_b$, $S_{ac}$, and $S_{bc}$ appear to be similar (Table 2).

### 3.1.2  Flow regimes characterized by the use of reservoirs

Kabini (C) is the only reservoir used for hydropower. Figure 7 shows that scenario $S_c$ generates a mean annual flow

that is the second highest, after that of $S_0$ with no reservoir in the basin. The mean annual flows of combined irrigation

and hydropower reservoirs ($S_{ac}$ and $S_{bc}$) are higher (1,076-1,289 $m^3 s^{-1}$) when compared with that of two irrigation

reservoirs ($S_{ab}$). Similarly, the mean annual flow of scenario $S_{abc}$ with three reservoirs is around 906 $m^3 s^{-1}$, which is

more than those of the scenarios $S_{bd}, S_{cd}, S_{ad}$ but less than those of $S_{bc}, S_{ab}$ and $S_{ac}$ with two reservoirs. This is because

Kabini (C) is a hydropower reservoir that does not divert water from the river, but releases water frequently and

ensures flow above a certain threshold resulting in a higher mean. The comparison of a scenario of configuration with

two irrigation reservoirs and one hydropower reservoir ($S_{abc}$) to a scenario with two irrigation reservoirs ($S_{bd}$) indicates

that the former has less impact on mean annual extreme flow conditions such as 1, 2 and 7-day minimum than the

latter (Figure 7).

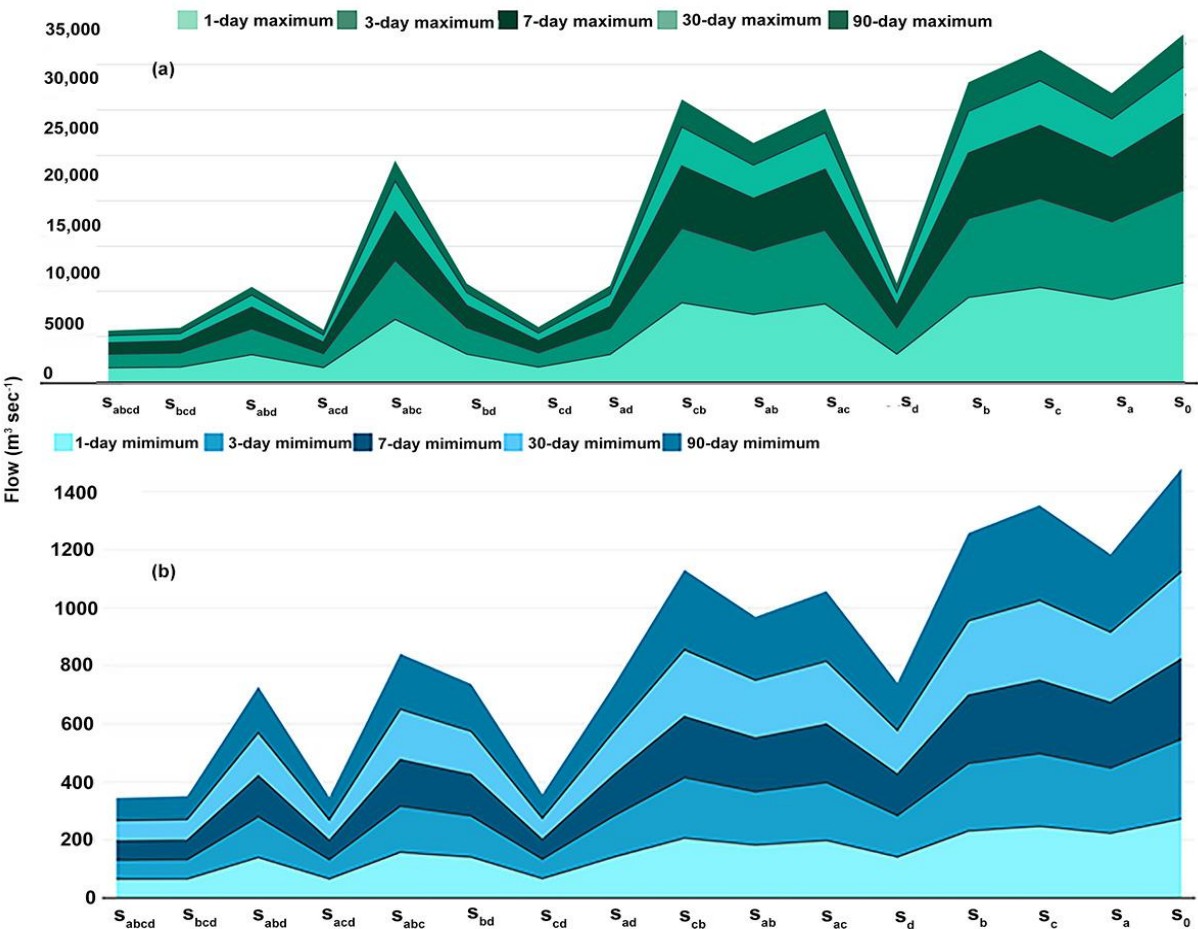


**Figure 7. The magnitude of annual extreme flow conditions (upper panel: maxima, lower panel: minima) of flow regimes generated by different configurations of reservoirs**

Comparing similar configurations in Figure 7 of two reservoirs only for irrigation ($S_{ad}$ and $S_{bd}$) versus those that contain the hydropower reservoir ($S_{cd}$) indicates that the hydropower reservoir decreases the low pulse count and low pulse duration compared to irrigation reservoir.

Table 2. Overview of duration and environmental flow parameters of IHA for different scenarios of reservoir configurations.

| Scenarios | Hydrological impact parameters | | | | | Environmental flow parameters ($m^3\,s^{-1}$) | |
|---|---|---|---|---|---|---|---|
| | Mean annual flow ($m^3/s$) | Low pulse count (days) | High pulse count (days) | Low pulse duration (days) | High pulse duration (days) | Extreme low peak | Extreme low frequency |
| Scenario with four reservoirs Integrated | | | | | | | |
| $S_{abcd}$ | 265 | 2.2 | 3.4 | 52.5 | 16.6 | 44.9 | 1 |
| Scenario with three reservoirs integrated | | | | | | | |
| $S_{bcd}$ | 296 | 2.4 | 3.6 | 44.1 | 73.3 | 44.9 | 0.9 |
| $S_{abd}$ | 443 | 1.4 | 3.9 | 90.5 | 16.8 | 66.9 | 0.9 |
| $S_{acd}$ | 274 | 2.6 | 3.6 | 46.3 | 29.1 | 44.9 | 1 |
| $S_{abc}$ | 907 | 2.3 | 3.8 | 57.7 | 17.1 | 117 | 1.4 |
| Scenario with two reservoirs integrated | | | | | | | |
| $S_{bd}$ | 480 | 1.8 | 3.9 | 75.6 | 88.3 | 61 | 0.6 |
| $S_{cd}$ | 310 | 2.2 | 3.5 | 55.9 | 79.7 | 48.7 | 1.1 |
| $S_{ad}$ | 452 | 1.4 | 4.2 | 90.2 | 89.8 | 67.1 | 1 |
| $S_{bc}$ | 1289 | 2.6 | 3.8 | 46.4 | 29.4 | 181.1 | 1.6 |
| $S_{ab}$ | 995 | 1.6 | 3.1 | 86.9 | 17.1 | 119.1 | 1.4 |
| $S_{ac}$ | 1076 | 2.9 | 3.6 | 46.8 | 29.9 | 181 | 1.7 |
| Scenario with one reservoir Integrated | | | | | | | |
| $S_d$ | 488 | 1.9 | 4 | 74.2 | 91.9 | 60.8 | 0.6 |
| $S_b$ | 1377 | 2.6 | 3.6 | 42.5 | 103.7 | 181.9 | 1.6 |
| $S_c$ | 1460 | 2.4 | 3.5 | 42 | 95.7 | 242.9 | 2.1 |
| $S_a$ | 1164 | 2.6 | 3.4 | 48 | 29.9 | 182.8 | 1.4 |
| Scenario with no reservoir | | | | | | | |
| $S_0$ | 1548 | 2.4 | 4 | 45 | 109.7 | 242.9 | 2.1 |

### 3.1.3  Flow regimes characterised by varying the configuration of reservoirs

Harangi (A) and Hemavathi (B) reservoirs are located in the upstream area of the basin, on one of the tributaries of the Upper Cauvery. Harangi (A) reservoir is the smallest in terms of volume, followed by Kabini (C), Hemavathi (B), and KRS (D). When comparing the flow altered by configurations with only one reservoir, $S_a$ produces regimes with lower mean annual flows than $S_b$. Generally, reservoirs with longer residence times tend to have larger impact on the flow regimes compared to reservoirs with smaller residence times (see Supplementary materials, Table S.5 for residence times). However, $S_a$ (with Harangi reservoir) has higher impact on the flow regime than $S_b$ (with Hemavathi reservoir). One reason could be that M.H. Halli sub-basin (with Hemavathi reservoir with a large residence time) receives the highest rainfall compared to other regions in the Upper Cauvery (Reddy et al., 2023), which contributes towards a lower impact of $S_b$ compared to $S_a$.

Furthermore, in the absence of its reservoirs, the mean annual flow in M.H. Halli sub-basin is lower (75 $m^3$ $s^{-1}$) when compared to Kudige (139 $m^3$ $s^{-1}$), T. Narasipur (349 $m^3$ $s^{-1}$) and Kollegal sub-basins (630 $m^3$ $s^{-1}$). This shows that M.H. Halli sub-basin contributes little to the overall flow. As a result, the $S_a$ scenario generates a lower mean annual flow than the $S_b$ scenario. Similarly, for two reservoirs configurations, the M.H. Halli sub-basin has a lower no-reservoir mean flow than the Kudige sub-basin. As a result, $S_{ac}$ performs worse than $S_{cb}$. Among the configurations with three reservoirs, the mean annual flow and other indicators of hydrological alterations of the $S_{bcd}$ and $S_{acd}$ scenarios were as undesirable as the four-reservoir scenario. It is acknowledged that $S_0$, being the unregulated scenario without any reservoir, exhibits the highest flow due to the absence of flow regulation and water diversion. In contrast, $S_c$, which is a configuration with only a hydropower reservoir, needs to release water regularly for electricity generation purposes. As a result, $S_0$ is estimated to have the highest mean annual flow, followed by $S_c$ and $S_b$.

Since the configuration $S_{abd}$ has Hemavathi reservoir which falls in the M.H. Halli sub-basin that receives highest rainfall, thereby contributing significantly to the overall flow, $S_{abd}$ has less impact on flow regime compared to $S_{acd}$ despite Kabini (C) having less storage capacity compared to the Hemavathi reservoir (B).

### 3.2 Agricultural production

The agricultural production in the sub-basins is calculated based on the assumption that irrigated area becomes unirrigated (*i.e.* rainfed) when the corresponding reservoir is removed in a spatial configuration scenario, without changing the crops that are being cultivated. The proportion of cultivated and irrigated land is given in Figure 8. Figure 9 shows the economic values of various crops grown in each of the four sub-basins, based on the flow regimes simulated by the integrated model with and without its respective reservoirs. In Figure 9, each sub-basin is studied one at a time to demonstrate the economic value of irrigated crop cultivation.

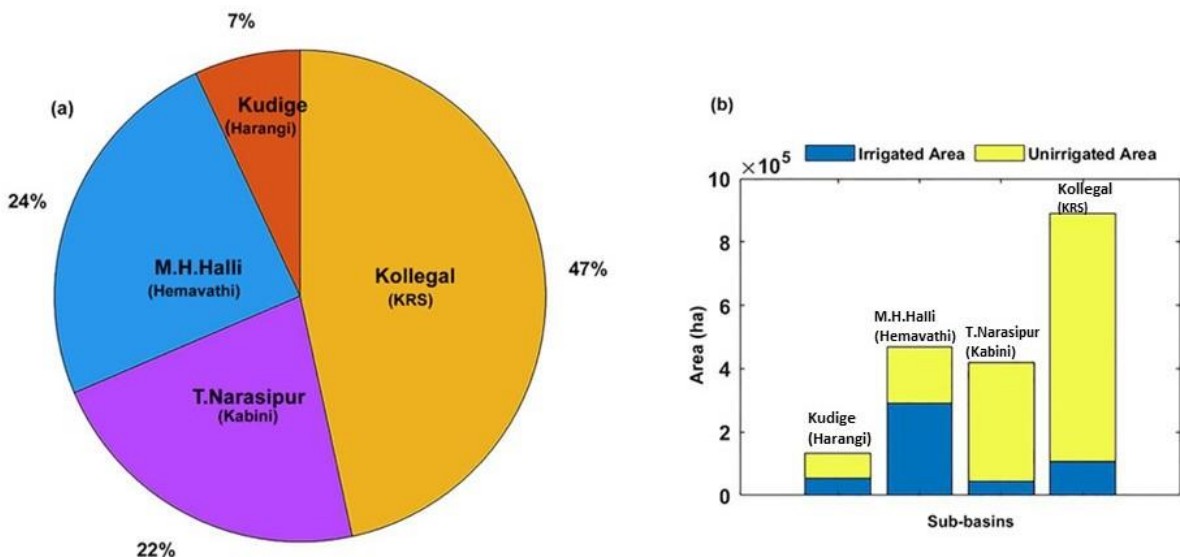

Figure 8. Overview of cultivated areas in different sub-basins. (a) the contribution of sub-basins to the total cultivated area of the Upper Cauvery basin, and (b) the irrigated and unirrigated (or rainfed) areas in each sub-basin

Five categories of crops were distinguished, namely, cereals, pulses, oilseeds, horticultural & plantation (H&P) crops, and spices. Among horticultural & plantation crops, coffee, coconut and cashew nut contributed 65 percent of the total H&P cultivated area (Figure 9, author's estimation). According to current estimates, the contribution of plantation crops accounts for 58 percent of the economic value of the H&P crops (see Figure 9, author's estimation).

Figure 9 shows that the horticultural crops and spices contributed most to the economic value in all sub-basins. In M.H. Halli and Kollegal sub-basins, where the area under cereals is high, the economic value of cereal production is low compared to that of the horticultural crops and spices.

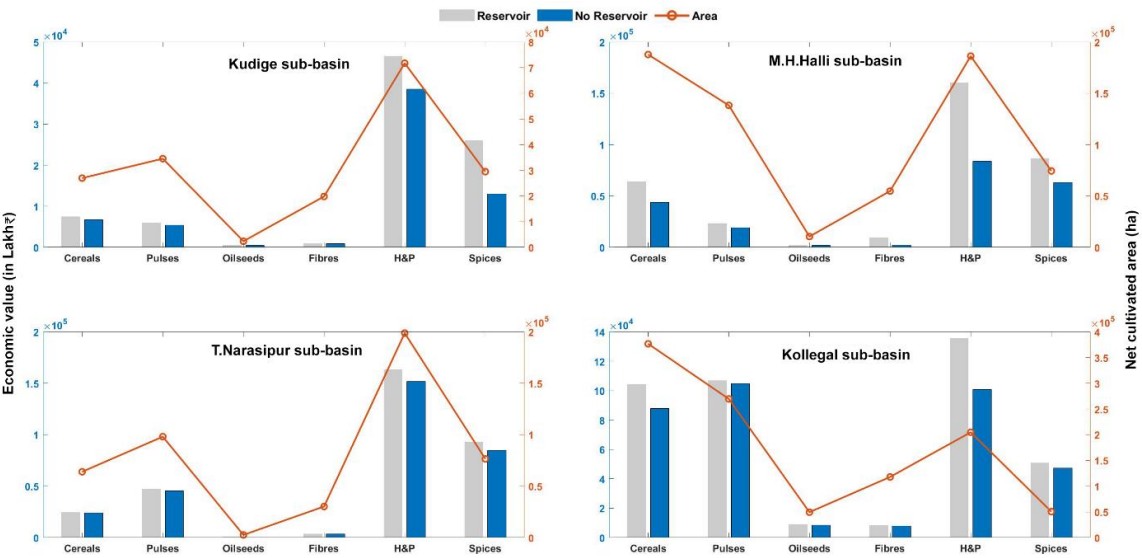

Figure 9. The economic value (Lakh ₹ per year; 1 Lakh = 100,000) of different crop groups in individual sub-basins, with and without its respective reservoirs.

When comparing the economic value of crops within a sub-basin with and without its reservoir, not much difference was observed in the economic values of pulses, oilseeds, and fibres in all the sub-basins. The differences in economic values with and without its reservoir are significant among horticultural crops and spices in three sub-basins, *i.e.* Kudige (Harangi), M.H. Halli (Hemavathi) and T. Narasipur (Kabini) sub-basins. In Kollegal (KRS) sub-basin, the majority of crops are rainfed and only 10 percent is irrigated, which explains the small difference in the economic value with and without its reservoir.

The values generated by alternative dam planning and design scenarios in comparison to the existing reservoirs as the baseline can be studied by varying the spatial configurations of the reservoirs (Figure 10). It demonstrates how economic value from agricultural production varies across the various scenarios of reservoir configurations. In general, increasing the number of dams does raise the economic value of agricultural production as compared to scenario $S_0$ (without any dams). The presence of all four dams in the basin generates the highest economic value from the agricultural production. Note that the agricultural value of $S_0$ (no dams and therefore also no irrigation) is approximately 67 % of the present situation, $S_{abcd}$, with irrigation in command areas of the four reservoirs.

The scenario of four dams $S_{abcd}$ does not show a dramatic increase in value as compared to the scenarios of the configurations with three dams. Among the scenarios with two dams, there are three configurations, *i.e.* $S_{bd}$, $S_{bc}$, and $S_{ab}$, that show much higher value generation than other scenarios of configurations with two dams and are comparable to the scenarios with three and four dams. In the case of scenarios with one dam, scenario $S_b$ shows a much higher economic value generation. This is because the Hemavathi reservoir (B) has a well-developed command area growing mainly horticultural crops that fetch high prices.

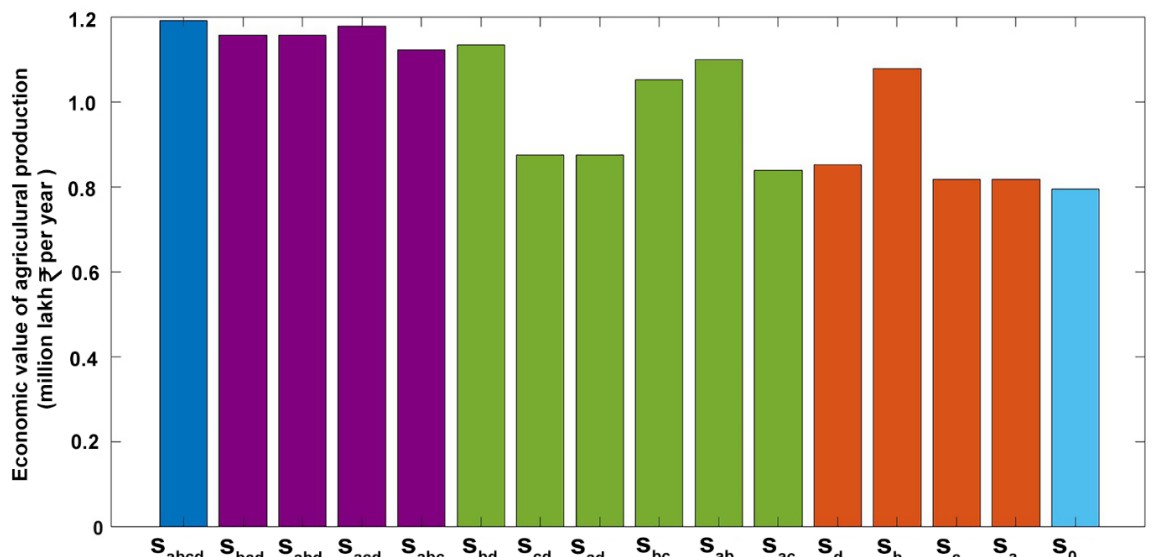

Figure 10. The economic value of agricultural production under different scenarios of spatial configurations of the reservoirs

### 3.3 The normalized fish diversity index across sub-basins

The Fish Species Richness (FSR) value is derived based on a global statistical model developed by Iwasaki et al. (2012), which is then converted into a normalized fish diversity index (NFDI). The results of normalized fish diversity index (NFDI) calculations for different spatial configurations of the reservoirs are shown in Figure 11, which ranges from 0.25 to 1.00 The values obtained by Iwasaki et al. (2012) are in the range of 20 to 250 species (0.8 to 1.00 based on the normalized index). Other field studies have confirmed that the FSR in the Cauvery River Basin tends to be around 146 species (Koushlesh et al., 2021). Figure 11 also shows the mean annual flows for the various configurations.

The NFDI is greatly impacted by the configurations that contain a large reservoir (such as KRS) due to significant decrease in mean annual flow and in the coefficient of variations of low flow frequencies. This can be seen in the configurations containing

one ($S_d$), two ($S_{bd}$, $S_{cd}$, $S_{ad}$) and three ($S_{bcd}$ , $S_{abd}$ ,$S_{acd}$ ) reservoirs where lower NFDI values are observed. Among the scenarios of configurations with two reservoirs, $S_{ad}$ has better NFDI than $S_{bd}$ despite having lower mean annual discharge, demonstrating the effect of other hydrological flow regime parameters on NFDI. Among the configurations containing three reservoirs, not much difference in NFDI values are observed except in $S_{abc}$, which scores higher than other configurations containing three

reservoirs ($S_{bcd}$, $S_{sbd}$ and $S_{acd}$). These latter configurations contain KRS, which is the most downstream and the largest reservoir and include two smaller reservoirs out of three in various spatial configurations upstream of the KRS reservoir. This shows that a very large reservoir can dominate the effect of reservoirs on the flow regime characteristics and consequently on NFDI.

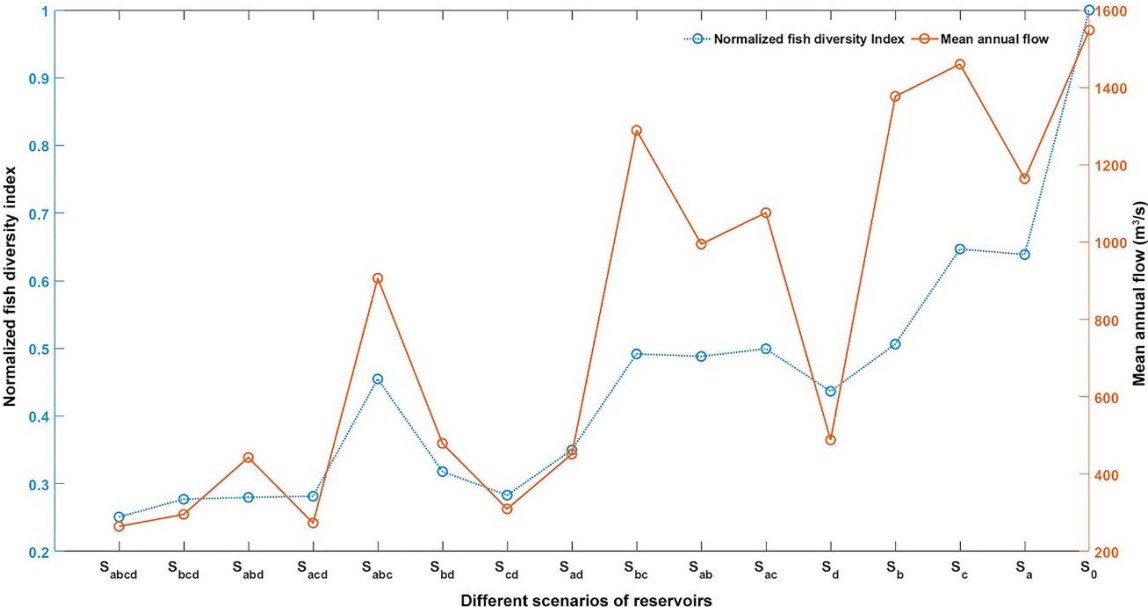

Figure 11. The normalized fish diversity index (NFDI) of the different configurations of reservoirs was calculated based on

mean discharge and flow regime characteristics.

### 3.4 The production possibility frontier (PPF)

The agricultural production and the normalized fish diversity index (NFDI) for different spatial configurations of the reservoirs define the convex hull of the production set. (Figure 12). The Production Possibility Frontier (PPF) is then defined as the outward boundary of the production set. The points and the corresponding configurations lying on this boundary are deemed to be more desirable than the points lying inside because the ecosystem services linked to agricultural production and NFDI are provided less efficiently by the bioeconomy of the basin in the case of the latter than the former.

The findings show that the scenario without any reservoir ($S_0$) is advantageous for the fish species through the lens of the normalized fish diversity index (NFDI) used in this study. Due to lower values from agricultural production, scenarios of configurations with one reservoir ($S_d$, $S_a$ and $S_c$) and two reservoirs ($S_{cd}$, $S_{ad}$, and $S_{ac}$) perform poorly with respect to the frontier. However, due to lower values of NFDI, scenarios of configurations with four reservoirs ($S_{abcd}$), three reservoirs ($S_{bcd}$, $S_{abd}$, $S_{acd}$) and two reservoirs ($S_{bd}$ and $S_{bc}$) are also considered inferior with respect to the frontier. The scenario $S_{bc}$ is however slightly

worse off in terms of NFDI and agricultural production, relative to the PPF.

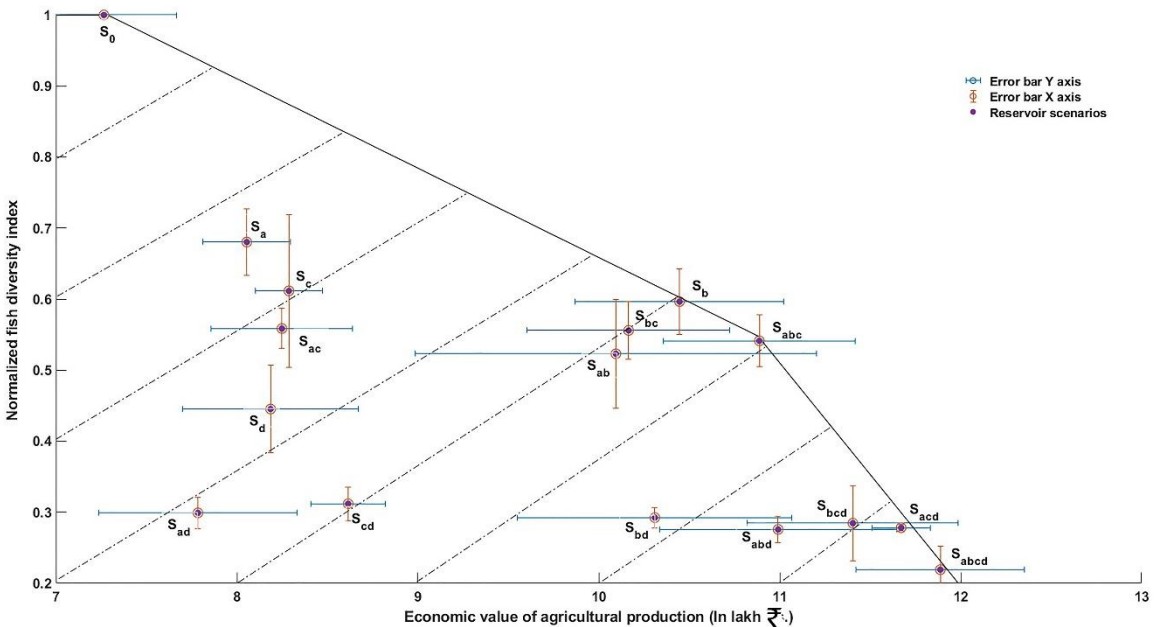

Figure 12. Illustration of production set and production possibility frontier (PPF). The PPF is the outer edge of the set, between the value of agricultural production and normalized fish diversity index (NFDI). The error bars around the mean values of agricultural production and NFDI, for a given reservoir scenario, are one standard deviations of the agricultural production and

NFDI simulations for the reservoir scenario over different years.

Five scenarios of configurations $S_0$, $S_b$, $S_{abc}$, $S_{acd}$, and $S_{abcd}$ define the frontier. The scenario of the configuration with all reservoirs ($S_{abcd}$) produces the highest value of agricultural output but has the lowest NFDI. The scenario $S_b$ is the only one with a single reservoir (Hemavathi reservoir B) that serves irrigated crops with a relatively good NFDI. Scenarios $S_b$, and $S_{abc}$

do not include the KRS (D) reservoir with the largest storage capacity, and thus the flow regime was not significantly altered as compared to the cases of $S_{abcd}$ and $S_{acd}$. This resulted in better NFDI for fish species and a better 'balance' between agricultural production value and NFDI. Finally, both $S_{abc}$ and $S_{acd}$ are on the frontier because the KRS (D) reservoir in the

scenario $S_{acd}$ adversely altered the flow regime by diverting more water for agriculture, thereby boosting agricultural production but simultaneously limiting the NFDI for fish species.


## 4. Discussion

### 4.1 Hydrological impacts of reservoirs on flow regime

The analysis of different combinations of reservoirs shows that the storage volumes of reservoirs have a significant impact on mean annual flows. For instance, a configuration adding a reservoir with high storage capacity and a large command area for

irrigated crops, such as KRS, leads to a notable decline in mean annual flow. Comparing scenarios with different combinations of irrigation and hydropower reservoirs it is observed that including a hydropower reservoir can mitigate mean annual extreme flow conditions by maintaining higher minimum flow levels during critical periods. However, it also highlights that the presence of a hydropower reservoir situated upstream of an irrigation reservoir may impact the frequency and duration of low flow pulses more than scenarios without hydropower reservoirs. These findings emphasize the importance of considering the

specific characteristics and objectives of different types of reservoirs when evaluating their impacts on the flow dynamics. The findings are consistent with a study conducted in the Lancang river in China where dams with storage capacities greater than 100 million $m^3$ had stronger impacts on streamflow regimes than smaller ones (Han et al., 2019).

Previous studies have indicated that hydropower dams cause monthly mean water levels to rise during the dry season and fall

during the wet season (e.g., Hecht et al., 2019). Even though the dry and wet seasons were not compared in the current study, we find that combining irrigation reservoirs with a hydropower dam has less impact on river flow regimes compared to combining reservoirs for irrigation purposes only. This is due to the regular water releases for energy production that maintain river flows year-round. The study also highlights that the reservoir-induced flow alterations can be compensated by tributary flow regimes. For example, the flow regime of a tributary can offset the low flow impact caused by a reservoir, resulting in a

lower overall impact on the flow regime downstream. Similar findings have been observed in other studies, where tributaries significantly contributed to controlling flooding in downstream areas (Pattison et al., 2014).

### 4.2 Social and ecological impacts

In the present study, the average contribution of a reservoir to agriculture production was estimated to be ₹ 0.40 billion per year ($ 0.005 billion per year[2]). It not only supports food security but also contributes to economic development and growth. Most of the horticultural crops and spices that are grown in the Upper Cauvery basin are exported to earn foreign currencies. Fishing is another important ecosystem service supported by the river flow. The economic value of both commercial and subsistence fishing of the Cauvery River is estimated to be ₹35.93 billion per year ($ 0.44 billion per year) (Pownkumar et al., 2022). While direct economic contribution of fisheries to human wellbeing is significantly lower than that of crop production, fish populations and species richness have a significant role in sustaining the river environment such as population dynamics down the food web (Carpenter et al., 1985). But the ecological importance of fisheries in maintaining ecosystem services and functioning, which is indirectly supported by fish species richness, is often ignored in river basin management decisions.

The primary objective of using normalized fish diversity index (NFDI) is therefore not to predict values for fish species richness, but rather to demonstrate how different configurations of existing reservoirs can lead to different (fish) biodiversity conditions in the long run (since we are using averages of these two variables over 16 years). By assessing these relationships, it becomes possible to identify the potential impacts of reservoir configurations on the long-run biodiversity and ecological stability of the river systems. The scenarios containing the largest reservoir (KRS; D) had significant negative impacts on FSR due to declines in mean annual flows and the coefficient of variation of the low flow frequencies. When comparing scenarios that contained the hydropower reservoir with scenarios containing only irrigation reservoirs, the NFDI values were higher in the former indicating that irrigation reservoirs more adversely alter the flow regimes with respect to NFDI. Further, in contrast to configurations with two reservoirs, there was a significant difference in the NFDI values amongst the scenarios of configurations containing three reservoirs due to greater alterations in flow characteristics.

In contrast, no significant difference in the economic value of agricultural production for different scenarios of configurations were observed based on storage volumes, the purpose of the reservoirs, and the orders of the streams on which the reservoirs are constructed. The economic value of agricultural production appears to be largely influenced by the area irrigated per unit volume of stored water in the reservoir. This means that if water is being stored for irrigation, then it should be used as efficiently as possible, *i.e.,* by producing high value agricultural products, to maximize value.

---

[2] 1 dollar equaled 81.66 rupees (₹) on 6 October, 2022.

**4.3 The role of PPFs in decision making**

The production set in Figure 12 shows the different configuration of two ecosystem services that can be produced using available water resources. The levels of ecosystem services that lie on the production possibility frontier (the outward boundary of the production set) represent the desirable production levels of the services. We limited our analysis to the existing set of reservoirs and did not synthetically include new reservoirs and the production set is defined as the convex hull of the 16 points. The construction of the convex hull is due to the discrete but realistic nature of the problem. There may be a continuum of production possibilities, but this continuum is neither real (because we only have the mentioned four reservoirs in the basin and therefore only 16 possible combinations of alternate realities depending on how these existing reservoirs could be removed in the future) nor within the scope of the current study. Given only a finite number of points, creating a convex hull to represent a convex production set, makes minimal assumptions and is consistent with the economics literature (see e.g., Ginsburgh and Keyzer, 1997). The latter (i.e., the inclusion of new reservoirs) might have provided us with a more exhaustive set of points, but this would have been impossible to validate.

The analysis based on the configurations lying on the PPF revealed that large dams that do not maximize the value of water stored, *i.e.*, by growing low value crops in smaller command areas, affect both NFDI and the economic value of agricultural production adversely. Such reservoirs are least favourable, as they are Pareto inferior to other configurations. In contrast, smaller reservoirs on tributaries (away from the main river stem) that grow high-value crops and maximize the value of water stored are Pareto superior and most preferred. Small reservoirs then significantly increase the value of the water while have a lower detrimental effect on areas upstream and downstream (Van der Zaag and Gupta, 2008). For decision-making, this means that large reservoirs that do not maximize the value of water stored should be discouraged and smaller more effective reservoirs should be encouraged if faced with a choice between the two types of reservoirs. However, larger reservoirs are substantially less expensive (per m³ of water storage capacity) than smaller reservoirs due to economies of scale, and as a result, the ecological costs must be included during the cost-benefit analysis (Van der Zaag and Gupta, 2008).

**4.4 Ecosystem service perspective on PPF and future challenges**

Understanding ecosystem service (ES) interactions was achieved through the interpretation of the production possibility frontier. However, the complexity of the interactions may prevent the translation of ES knowledge into decision-making processes (Vallet et al., 2018; Hegwood et al., 2022). In the present study, the scenario without reservoirs (S0) was

hydrologically a superior choice in terms of fish species richness. However, it had the lowest agricultural output, which would

negatively affect employment generation and economic growth. Similar to this, the integration of all four reservoirs (Sabcd) would boost agriculture production by increasing the area of land irrigated but at the expense of lower fish species richness that would be detrimental to riverine ecology. The combination Sb  and Sabc, which can enhance both ecosystem services, yield more balanced results.

However, intangible services were not analysed in this study. For example, humans directly consume or use both agriculture and fisheries products for food, nourishment, and employment, and to support their way of living. Both agroecosystems and fisheries provide regulating and supporting services that are crucial for ecosystem functioning and resilience. However, the human-driven ecosystem dis-service from agricultural activities can reduce ecosystem resilience and decrease service generation that are necessary for human survival. Therefore, the non-tangible ES and dis-services should also be taken into

consideration using appropriate economic valuation tools in a tradeoff analysis. Further, there is a need to determine which efficient ES combinations would be preferred by stakeholders by assessing indifference curves that describe human preferences for different ecosystem services including regulating and supporting services (Cavender-Bares et al., 2015; King, et al., 2015).

### 4.5  Limitations of the study and future work

The limitations of the presented work and areas of further research are now briefly discussed.

#### 4.5.1    Model assumptions and uncertainty

We acknowledge that no model is perfect. In the present study, the reservoirs operations at a daily scale are based on trigonometric functions that only incorporate water demand by various command areas as the dominant driver of reservoir

releases. Accommodating dam-specific water releases might improve the simulation of intra-monthly variability in streamflow (see its discussion in Ekka et al., 2022). Therefore, enhancing the model calibration process may involve incorporating operating rule curves that also consider specific reservoir functions and flow requirements. Whether this leads to changes in the conclusions drawn based on the possibility frontier shown in Figure 12 is beyond the current scope. However even if we assume log effects of mean annual flow on NFDI, changes in flows of one or two orders in log scale would not affect the

conclusions drawn (since NFDI is a function of log of mean flows and other streamflow characteristics). Hence, a reservoir configuration leading to substantial alterations in streamflow characteristics — deviating not just marginally but significantly

from mean flows — would profoundly impact the NFDI. It must demonstrate a substantial increase in economic value to remain a Pareto superior choice. Reservoirs that significantly alter flow regimes but do not add significant value should therefore be discouraged since it would be a Pareto inferior choice.


Further, although it is acknowledged that the current analysis does not directly provide a practical solution, it highlights an important consideration for reservoir planning and management. The paper presents a proof of concept of the trade-off between the economic benefits of existing reservoirs for agricultural production and the potential negative impacts on fish diversity. While using the normalized fish diversity index as an indicator, the study provides an assessment of change in some aspects of freshwater habitat integrity. We have applied the equation developed by Iwasaki et al. (2012) and Yoshikawa et al. (2014) to the Upper Cauvery basin and have extended the application of space and time substitution based on the equation (by time here we mean the occurrence of different scenarios). The central idea is to assess how environmental quality varies with different reservoir configurations and how it trades off with agricultural production. We acknowledge the limitation of equation 5 that in explaining the variability in normalized fish diversity index it does not consider other chemical and biological factors since it is solely based on the assessment of changes in water quantity and not quality, nor of impacts of non-dam related interventions. The same holds for our model. If the impact of unaccounted variability, e.g. of water quality and non-dam related interventions, on fish species richness (FSR) exceeds the recognized reservoir-induced streamflow variability, the reliability of changes in FSR values based on Equation 5 may be compromised. Unconsidered unknown variables like human footprint and fragmentation can introduce bias (Schipper and Barbarossa, 2022).


We cannot verify what NFDI values are for the hypothetical scenarios since there are no counterfactuals. However, the 'observed' NFDI around the gauge station where equation 5 is being used to assess the environmental quality of various scenarios via NFDI is around 0.20, which is close to the estimated value of 0.24 by equation 5 (for the current state as the scenario with all reservoirs in place - the only scenario that is factual). By using the normalized fish diversity index, our analysis also desensitizes the use of absolute numbers of FSR (and absolute changes) and thus focuses more on the relative rankings in the tradeoff space in terms of proxies of environmental quality and agriculture production. Therefore, the innovation indeed lies not in applying the same equation but in building on Iwasaki et al. (2012) and Yoshikawa et al. (2014) to apply their equation for various configurations of existing dams and how that is used in the tradeoff analysis. Also, Yoshikawa et al. (2014) provided a sensitivity analysis based on reducing flows of a certain basin by a certain percentage, and suggested

consideration of sensitivity analysis in future studies. The construction of our production possibility frontier in this regard can be seen as a sensitivity analysis where various combinations lead to scenarios of streamflow alterations due to dam regulation, irrigation, and other uses and how FSR based on equation 5 is sensitive to it. To keep the index of environmental quality (NFDI) comparable between the scenarios (where reservoirs are placed or removed in combinations upstream), we only applied the equation at the most downstream gauge. The use of NFDI is more of a means to assess the capacities to have certain levels

of fish diversities in various reservoir scenarios, assuming streamflow changes are the dominant effects – in the case of damming this means loss of diversity (see e.g. Zarfl et al., 2019; Ganassin et al., 2021), while the case of less dams leads to higher capacity and species recovery (see e.g. Bednarek, 2001; Hansen and Hayes, 2012).

  However, to address the limitations of the approach more effectively, further investigation and field information are required. To determine an appropriate threshold level of fish reduction, a comprehensive assessment of specific requirements of fish

habitats, their migration patterns, and population dynamics in the presence of reservoirs is needed. This involves studying factors such as water temperature, dissolved oxygen levels, substrate composition, and availability of food sources. Additionally, assessing the migration patterns of fish can help identify potential barriers created by reservoirs and develop mitigation measures to facilitate their movement. Furthermore, studying population dynamics will provide insights into how the presence of reservoirs affects fish reproduction, growth, and overall population size.


### 4.5.2     On dominant ecosystem services in the construction of PPF

The current analysis of the Production Possibility Frontier (PPF) does not include the consideration of riverine and culture fisheries in reservoirs. These fisheries are estimated to have an economic value of approximately $0.59 million per year, representing around 12 percent of the economic value of agricultural production ($5 million per year). Also, the economic

value generated by hydropower was not considered because only one of the four existing reservoirs supported it. Moreover, the study assumed that when an irrigated area is associated with a reservoir that is withdrawn, it becomes unirrigated (rainfed). This assumption may have influenced the economic value of different scenarios, as farmers might adjust their production practices in response to the change in irrigation. Future research can also consider synthetic reservoirs to more exhaustively explore alternative production sets and include values generated from multiple uses and changing cropping patterns.


While calculating the economic value of crops, the size of the cropped area is kept constant to isolate and analyze the impact of various reservoir combinations on economic outcomes. This approach simplifies the modeling process and helps in

understanding the relationships and interactions between varying reservoir combinations, and crop production, without the added complexity introduced by varying land sizes. This simplification, however, comes with limitations. For example, in the face of varying water allocations, farmers can adopt various strategies related to changing crops, for example changing to rain-fed agriculture or shifting towards less water-intensive irrigated crops (Graveline & Mérel, 2014).

Another limitation of this study is the utilization of constant prices, a factor that may pose challenges in assessing the impact of droughts and reduced water allocations on crop yields. If the basin is large enough and dominates the domestic market in terms of production of certain crops, then droughts and reduced water allocations will reduce crop yields, which will constrain supply and can therefore significantly affect prices. Since agricultural demand is highly inelastic, significant changes in supply may lead to abrupt changes in prices (Haqiqi et al., 2023, 2022; Parrado et al., 2019). As agricultural markets are well-developed in the basin and well-connected to other domestic and international markets outside the basin, production changes in the basin, could be compensated by production in  neighboring places unless there is a significant supply shock.

## 5    Conclusion

The main objective of the paper was to evaluate the hydrologic, ecological, and economic impacts of multiple existing dams in the Upper Cauvery River basin, India. To do so, a novel approach was presented that estimated the production of river ecosystem services using a landscape based hydrological model integrated with the modelling of the operations of multiple existing reservoirs at daily scale when pre-dam data is unavailable. The high resolution and robust simulation of pre-dam flow regimes offered the unique opportunity to assess the effects that cascades of existing reservoirs have on the river flow regimes downstream in a virtual experiment setting. Such a study has been conducted for the first time, especially for the case of Indian river basins where pre-dam data is unavailable but there are increasing calls for environmental impact assessment of large multiple dams (Erlewein, 2013; Lele, 2023).

The hydrological impacts of different configurations of reservoirs were assessed using Indicators of Hydrological Alterations. The biophysical quantification of major ecosystem services, indicated by the economic value of crop production and fish species richness, supported by the river were estimated and a production possibility frontier, representing the tradeoff between the two, was quantified. The main findings that can enhance our understanding of the effects of multiple existing dams on the provision of dominant ecosystem services and help optimize river management plans are summarized below.

- The mean annual flow and annual extreme conditions of minimum and maximum flows are adversely affected by the largest dam in terms of storage. In comparison to reservoirs used just for irrigation, scenarios of reservoirs used for hydropower and irrigation have less impact on low flow pulses and low flow duration.

- The large dam in the sample did not maximize the value of water stored. We found that low value irrigated crops were cultivated, which adversely affected both FSR and the economic value from agricultural production. Such a reservoir is the least favourable and should be discouraged by policy makers.

- Growing high value irrigated crops with a highly established command area served by small and medium reservoirs can strike a favourable balance between agricultural production and fish species diversity.

- Heavily altering the river landscape with reservoirs (e.g., by maximizing the number of reservoirs) provides a superior result in the sense that it maximizes agricultural income. However, it may not be preferred by diverse stakeholders such as fishers and environmentalists due to dismal biodiversity that it leads to, as indicated by fish species richness (FSR). Such an option produces lowest FSR. This perhaps should be favoured less than a configuration of reservoirs that strikes a favourable balance between agricultural production and fish species diversity while still efficiently producing both. This goal could also be achieved by prioritizing the enhancement of rainfed agricultural production. By doing so, we can potentially minimize the tradeoffs with other critical ecological services compared to irrigated agricultural production. By reducing the tradeoffs with other ecological services and enhancing water management practices, we can strive for a more sustainable and balanced approach to water resource management in the basin.

**Competing interests**: One of the co-authors is a member of the editorial board of the journal Hydrology and Earth System Sciences. The peer-review process was guided by an independent editor, and the authors have also no other competing interests to declare.

**Acknowledgements**

This research was financed by the Indian Council of Agricultural Research, Ministry of Agriculture Government of India through a scholarship for the PhD study [18(26)/2016-EQR/Edn] for the principal author. The authors are thankful to National Data Centre, Central Water Commission, New Delhi for providing gauge station data and GIS-related information on the Cauvery River basin. The authors wish to acknowledge the help provided in this research by the water efficiency task force officers under the India-European Union Water Partnership (IEWP).

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
