# Peer review of "How economically and environmentally viable are multiple dams in the Upper Cauvery basin, India? A hydro-economic analysis using a landscape-based hydrological model"

_Hydrology and Earth System Sciences, 2023_

## Author Response (AR1)

**Reviewer #1**

| Reviewer's Comment 1st round | |
|---|---|
| After reviewing the new preprint, I am concerned that several of the most important comments to the original submission have not been addressed. While the new version of the paper and the accompanying responses from the authors provide extensive explanations of the previous assumptions and rationale of the study, they, unfortunately, do not address some of the fundamental methodological flaws and misconceptions I identified in the initial review. | Thank you very much for taking the time to review the revised preprint. We take your feedback seriously and are committed to addressing these concerns comprehensively. We carefully reevaluated our methods and assumptions to ensure their accuracy and validity. Additionally, we revised the manuscript to provide clearer explanations and justifications where necessary. |
| In particular, in the new version of the paper, the authors have not provided a satisfactory response to the comments about the misuse of the fish species richness distribution concept. The new version insists on applying a global statistical model developed for the purpose of estimating the global distribution of fish biodiversity to predict local changes in biodiversity. I strongly believe that this is an unfortunate extrapolation and NOT appropriate, as the rationale and purpose of the original FSR model by Iwasaky et al. is NOT to explain the phenomenon the authors claim to study (i.e., how water reallocation to change the mean flow of a river reach over a few years could increase or decrease biodiversity in a basin). | Our revised manuscript clearly states that this paper is the first study of its kind to assess the tradeoff between two ecosystem services as indicated by income from agricultural production and fish species richness in the Indian context. This study sacrifices high accuracy in predicting fish species richness for modeling tradeoffs between agricultural production and fish species richness (FSR). The limited prediction accuracy has been acknowledged and it has been highlighted that this needs more field campaigns and left for future research.

**Kindly see lines 377-393 in the revised manuscript as provided below**

"In other words, flow characteristics other than mean river discharge also play a vital role in sustaining aquatic ecosystems. Many Species Discharge Relationship (SDR) models have been derived based on data of large basins (>500 km$^2$) globally to explain long-run riverine fish species richness (FSR) as a function of discharge and other variables (Schipper and Barbarossa, 2022; Xenopoulos and Lodge, 2006; Iwasaki et al., 2012). In the present study, the basin is >10,000 km$^2$, at which scale discharge is a key variable explaining differences in |

| | species richness (Schipper and Barbarossa, 2022, page 1502). We adopted an empirical function (equation 5) by Iwasaki et al. (2012) to quantify fish species richness. We use the equation to assess changes in fish species due to changes in flow characteristics for the same basin (keeping area and latitude constant to incorporate the fixed effect of the basin). This is very similar to the use of the Budyko curve derived from basin data sets across the globe in hydrology, e.g. space for time substitution to assess the impacts of changes in precipitation on rainfall partitioning in basins in the long run (Bouaziz et al., 2022). Indicators for flow characteristics, such as coefficient of variation of mean frequency of low flow in a year, coefficient of variation in the Julian date of annual minimum flow, and maximum proportion of the year in which floods have occurred, are also used. Here floods are defined as events when flows are greater than or equal to flows with a 60 % exceedance probability (Olden and Poff, 2003). This choice of a regression equation (equation 5) was suitable for our analysis since the underlying model does not consider water quality and other aspects" |
|---|---|
| Furthermore, in their response, the authors provide a "validation" of the results of the FSR model. However, the original model predictions range from 20 to 250 (~1200% range of variance), which in practice demonstrates that the model's predictive power is basically null: this range basically represents most medium-sized basins on the planet. | We agree that we should not use the term validation and it was an unfortunate use of the term. We have refrained from its use and limit ourselves to the use of the empirical equation. |
| The new version of the paper also doesn't provide a satisfactory answer to the extrapolation of the shape of the Pareto frontier of the PPF, assuming that there is a potential continuum and ignoring the discrete nature of the decision problem. | We recognize the significance of properly addressing the extrapolation of the Pareto frontier, particularly in the context of the discrete nature of the decision problem and therefore a suitable justification is provided in the **lines 666-675 of the revised manuscript** as indicated below: |

| | "The levels of ecosystem services that lie on the production possibility frontier (the outward boundary of the production set) represent the desirable production levels of the services. We limited our analysis to the existing set of reservoirs and did not synthetically include new reservoirs and the production set is defined as the convex hull of the 16 points. The construction of the convex hull is due to the discrete but realistic nature of the problem. There may be a continuum of production possibilities, but this continuum is neither real (because we only have the mentioned four reservoirs in the basin and therefore only 16 possible combinations of alternate realities depending on how these existing reservoirs could be removed in the future) nor within the scope of the current study. Given only a finite number of points, creating a convex hull to represent a convex production set, makes minimal assumptions and is consistent with the economics literature (see e.g., Ginsburgh and Keyzer, 1997). The latter (i.e., the inclusion of new reservoirs) might have provided us with a more exhaustive set of points, but this would have been impossible to validate". |
|---|---|
| It is also important to emphasize that with the clarifications of the new version of the paper, a large part of the manuscript consists of results related to the hydrological model development already published in Ekka et al. 2022. A simple citation and quotation of the previous results might be sufficient instead of reproducing the text here. | The modelling additions we have made are in response to the referee comments which we believe is fair also for this paper to be complete on its own. A citation and further explanation is provided.

**See lines 430-437 of the revised manuscript as changes indicated below**

"This section first reports on the quality of the model developed for the study area. Table S.3 reports on the calibration and validation performance of the model developed for the study area in Ekka et al. (2022). The model was calibrated using the NSGA II multi-objective optimization algorithm, and the Pareto front ranges for both -NSE (Nash Sutcliffe Efficiency) and MAE (Mean Absolute Error) are shown (see supplementary materials for more details). Note the use of negative NSE was used alongside MAE to minimize the two objective |

| | functions jointly (and therefore maximize NSE alongside minimization of MAE). The developed model is then used to simulate flow regimes for the 16 scenarios of different spatial configurations of existing reservoirs as shown in Table 1, and the degree of hydrological alterations is assessed". |
|---|---|
| In some cases, the answers given are also contradictory. In response to the comment of the difficulty of applying a method like IHA to estimate flow regime alterations given the potential uncertainty and errors in modeling the system, the authors claim that the model is adequate. However, the limitations sections of the revised manuscript claims that "It is acknowledged that specific water releases from specific dams may not have been accurately captured by the reservoir operations model". so how can hydrologic alteration be correctly assessed if operations are not accurately captured? | Research inevitably encounters limitations, yet these obstacles serve as opportunities for refinement leading to enhanced understanding and advancement in the field.

**The following text is added in the lines 711-722 of the revised manuscript**

"We acknowledge that no model is perfect. In the present study, the reservoirs operations at a daily scale are based on trigonometric functions that only incorporate water demand by various command areas as the dominant driver of reservoir releases. Accommodating dam-specific water releases might improve the simulation of intra-monthly variability in streamflow (see its discussion in Ekka et al., 2022). Therefore, enhancing the model calibration process may involve incorporating operating rule curves that also consider specific reservoir functions and flow requirements. Whether this leads to changes in the conclusions drawn based on the possibility frontier shown in Figure 12 is beyond the current scope. However, even if we assume log effects of mean annual flow on NFDI, changes in flows of one or two orders in log scale would not affect the conclusions drawn (since NFDI is a function of log of mean flows and other streamflow characteristics). Hence, a reservoir configuration leading to substantial alterations in streamflow characteristics-deviating not just marginally but significantly from mean flows-would profoundly impact the NFDI. It must demonstrate a substantial increase in economic value to remain a Pareto superior choice. Reservoirs that significantly alter flow |

| | regimes but do not add significant value should therefore be discouraged since it would be a Pareto inferior choice". |
|---|---|
| As a result, my assessment is that the new version of the manuscript should be rejected. In any case, I hope that the authors can make use of these comments to perform a comprehensive revision of critical components of their study.

Thank you for the opportunity to discuss the comments in this interactive session. Please see below our first response to the referee comments. We are looking forward to a fair and constructive discussion on the merit of this manuscript by further clarifying and responding to subsequent comments of the referee. | We appreciate the opportunity to engage in this interactive session and to discuss the comments raised by you. Your constructive feedback is instrumental in guiding us toward improvements in our work, and we are grateful for your dedication to ensuring the rigor and integrity of the scientific process.

We hope that we have provided sufficient explanation and made necessary modifications based on the feedback provided |

**References**

Schipper, A. M. & Barbarossa, V. (2022). Global congruence of riverine fish species richness and human presence. Global Ecology and Biogeography, 31, 1501–1512. https://doi.org/10.1111/geb.13519

Ginsburgh, V. I. C. T. O., & Keyzer, M. A. (1997). The structure of applied general equilibrium models. The MIT Press.

| Reviewer's Comment 2st round | |
|---|---|
| The authors claim that this is a first-time assessment, and that low precision is acceptable for such a study. While this may be a valid point, it actually distracts from the central point of the comment. I must clarify that the issue is not the precision of the study, but rather the fundamental epistemological inadequacy of the proposed approach. My central point is that a model developed to assess variability in biodiversity is not equivalent to a predictive model to explore causality between processes (such as how changes in flows can **cause** changes in biodiversity). | Our manuscript has highlighted that it provides a proof of concept of applying a novel methodology to analyze a tradeoff between two dominant ecosystem services in Upper Cauvery under plausible changes in reservoir configurations. As also stated in our response to Referee 3, the novelty is the tradeoff analysis that is unique to India based on model simulations that incorporate model simulations at a daily scale. |

| | |
|---|---|
| |

"The paper presents a proof of concept of the trade-off between the economic benefits of existing reservoirs for agricultural production and the potential negative impacts on fish diversity. While using the normalized fish diversity index as an indicator, the study provides an assessment of change in some aspects of freshwater habitat integrity" |
| The results presented highlight the unreasonableness of the proposed approach. According to the manuscript, the basin should have already experienced the extinction of ~100 fish species due to the allocation of water for irrigation, however, no evidence of such an ecological disaster is presented. Conversely, if some reservoirs are removed, will numerous fish species appear in this basin? This may sound like a caricature, but unfortunately, it is at the heart of the analysis presented. Again, this is a clear misuse of a regression model and the published literature on SDR and SAR; it is fundamentally a failure to distinguish between correlation and causation. | We appreciate your emphasis on the importance of distinguishing between correlation and causation in regression modeling, as well as the need for a careful consideration of the existing literature on Species-Area Relationship (SAR) and Species Distance Relationship (SDR) and therefore suitable justification with references is provided to strengthens the argument.

"Many Species Discharge Relationship (SDR) models have been derived based on data of large basins (>500 km$^2$) globally to explain long-run riverine fish species richness (FSR) as a function of discharge and other variables (Schipper and Barbarossa, 2022; Xenopoulos and Lodge, 2006; Iwasaki et al., 2012). In the present study, the basin is >10,000 km$^2$, at which scale discharge is a key variable explaining differences in species richness (Schipper and Barbarossa, 2022, page 1502). We adopted an empirical function (equation 5) by Iwasaki et al. (2012) to quantify fish species richness. We use the equation to assess changes in fish species due to changes in flow characteristics for the same basin (keeping area and latitude constant to incorporate the fixed effect of the basin). This is very similar to the use of the Budyko curve derived from basin data sets across the globe in hydrology, e.g. space for time substitution to assess the impacts of changes in precipitation on rainfall partitioning in basins in the long run (Bouaziz et al., 2022)" |

| | |
|---|---|
| As it stands, the study provides an assessment of change in some aspects of freshwater habitat integrity, not fish biodiversity. I'd respectfully encourage the authors to consider declining the current submission and reformulating the paper to explore the trade-offs in those terms | We understand and respect the sensitivity of the referee to the way we have used FSR in our manuscript. For our scale of study, FSR serves as an indicator of an important ecological service that quantifies environmental quality as seen through the lens of biodiversity and how its provisioning varies with more or less dams in the same basin. As also suggested by the referee, we have therefore rescaled the FSR to define an index between 0 and 1.

**The following modification is done as indicated in the lines 409-421 of the revised manuscript.**

"The fish species richness index is then normalized into an index, called the "normalized fish diversity index" ($NFDI_i$), for any $i^{th}$ scenario calculated as:

$$NFDI_i = \frac{FSR_i}{\max_i\{FSR_1,..,FSR_i,..,FSR_I)}$$

Where,
$NFDI_i$ is the Normalized Fish Diversity Index for the $i_{th}$ scenario
$FSR_i$ is the Fish Species Richness for the $i_{th}$ scenario
$I = 16$ is the number of scenarios of possible reservoir combinations (counterfactuals)

Utilizing the normalized fish diversity index in our analysis helps reduce dependence on absolute FSR numbers and their changes over different scenarios. Rather than focusing solely on numerical values, our methodology prioritizes the relative ranking within the tradeoff space. By incorporating proxies for environmental quality and agriculture, this normalization approach facilitates a nuanced assessment. It highlights the relative positions of various scenarios, providing insight into their impacts on both environmental quality and agricultural production". |

.

References:

Bednarek, A (2001). Undamming Rivers: A Review of the Ecological Impacts of Dam Removal. Environmental Management 27, 803–814. https://doi.org/10.1007/s002670010189

Bouaziz, L. J. E., Aalbers, E. E., Weerts, A. H., Hegnauer, M., Buiteveld, H., Lammersen, R., Stam, J., Sprokkereef, E., Savenije, H. H. G., and Hrachowitz, M. (2022). Ecosystem adaptation to climate change: the sensitivity of hydrological predictions to time-dynamic model parameters, Hydrol. Earth Syst. Sci., 26, 1295–1318, https://doi.org/10.5194/hess-26-1295-2022.

Ganassin, M J M, Muñoz-Mas, R, de Oliveira F J M, Muniz, C M, dos Santos, N C L, García-Berthou, E, Gomes, L  C (2021). Effects of reservoir cascades on diversity, distribution, and abundance of fish assemblages in three Neotropical basins, Science of The Total Environment, Volume 778, https://doi.org/10.1016/j.scitotenv.2021.146246.

Hansen JF, Hayes DB (2012) Long-term implications of dam removal for macroinvertebrate communities in Michigan and Wisconsin rivers, United States. River Research and Applications 28: 1540–1550.

Schipper, A. M. & Barbarossa, V. (2022). Global congruence of riverine fish species richness and human presence. Global Ecology and Biogeography, 31, 1501–1512. https://doi.org/10.1111/geb.13519.

Zarfl C, Berlekamp J, He F, Jähnig SC, Darwall W, Tockner K (2019). Future large hydropower dams impact global freshwater megafauna. Sci Rep. Dec 6;9(1):18531. doi: 10.1038/s41598-019-54980-8. PMID: 31811208; PMCID: PMC6898151.

| Reviewer's Comment 3rd round | |
|---|---|
| I appreciate your effort to develop an argument for your perspective on why it is appropriate and sufficient to use the SDR approach to model the effects of local flow changes on fish biodiversity and to constructively provide some options. | In our revised manuscript we have highlighted that it provides a proof of concept of applying a novel methodology to analyze a tradeoff between two dominant ecosystem services in Upper Cauvery under plausible changes in reservoir configurations. Further, we have discussed the limitations such as that the equation used for FSR is a statistical one and does not consider other chemical and biological |

I also understand the authors' urge to find a way to extend this study to the "environmental" domain: otherwise, there would be no point in claiming in the title that this is an "economic and environmental" assessment: there wouldn't be two axes of tradeoff analysis. However, since it is treated as a central part of the work, it needs to be rigorously revised. I cannot accept the authors' argument that this paper is not about FSR, or that this discussion is about "the reviewer's sensitivity to the way we used FSR in our manuscript". On the contrary, scientific novelty, merit, and rigor must be appropriately addressed following the authors' decision to use biodiversity as the "environmental" component of their study.

factors since it is solely based on the assessment of changes in water quantity and not quality and not of impacts of other non-dam related interventions, amongst other discussion points below.

**Kindly see lines 731-737 of the revised manuscript as indicated below**

"We acknowledge the limitation of equation 5 that in explaining the variability in normalized fish diversity index it does not consider other chemical and biological factors since it is solely based on the assessment of changes in water quantity and not quality, nor of impacts of non-dam related interventions. The same holds for our model. If the impact of unaccounted variability, e.g. of water quality and non-dam related interventions, on fish species richness (FSR) exceeds the recognized reservoir-induced streamflow variability, the reliability of changes in FSR values based on Equation 5 may be compromised. Unconsidered unknown variables like human footprint and fragmentation can introduce bias (Schipper and Barbarossa, 2022)"

I find your reference to Budiko fortunate. It may be a good example to help clarify the difference between correlation and causation. Let's do a mental experiment: A researcher, using the available global data of actual evapotranspiration (ETa) and runoff from energy-limited basins (that is, where PET < precipitation), performs a regression between these two variables and finds that there is a significant correlation of these fluxes, and publishes the results about this correlation. This is indeed a valid correlation (not spurious like earthquakes and solar flares) showing that around the world, basins with higher runoff tend to have more ETa.

Later, another study in a single basin where reduced streamflow has been observed in a river reach attempts to estimate changes in ET in that basin. These authors find the previously published regression model and use it to estimate the basin's reduction in ETa. The authors argue that this is a valid scientific result because there is a proven global correlation between these two fluxes. They also propose an "ET-R informed index" to assess the change in ET in this basin.

As you are probably aware, based on the understanding of the mechanisms that determine the interaction of water budget components, a decrease in runoff may be the result of increased ET at local scales for instances water is used in irrigation, the opposite of what the correlation implies. It is therefore incorrect to assume that because there is a correlation between ET and runoff across geographic areas there is an implied causality, and more importantly, that the direction of the correlation stands outside the domain of the original analysis.

This hypothetical scenario is basically analogous to the case the authors make. Your assumption of using an SDR equation to assess

Our methodology carefully applies global-scale equations to interpolate FSR variability within known data bounds, emphasizing large time/spatial scales. We acknowledge limitations regarding unaccounted variables' potential bias and inability to assess chemical/biological factors. Despite these, the consistent relationship between streamflow and FSR remains unchanged. While unable to verify hypothetical scenarios' FSR values, our analysis, utilizing relative indices, focuses on scenario ranking rather than absolute values. Therefore, we refrain from attributing positive influences of reservoirs on fish biodiversity in the Cauvery basin, aligning with our methodology's assumptions on generating counterfactual scenarios.

Therefore, we have revised the paper by discussing the limitations into different sections

4.5.1 model assumptions and uncertainty **(lines 710-762)**

"Model assumptions and uncertainty
We acknowledge that no model is perfect. In the present study, the reservoirs operations at a daily scale are based on trigonometric functions that only incorporate water demand by various command areas as the dominant driver of reservoir releases. Accommodating dam-specific water releases might improve the simulation of intra-monthly variability in streamflow (see its discussion in Ekka et al., 2022). Therefore, enhancing the model calibration process may involve incorporating operating rule curves that also consider specific reservoir functions and flow requirements. Whether this leads to changes in the conclusions drawn based on the possibility frontier shown in Figure 12 is beyond the current scope. However even if we assume log effects of mean annual flow on NFDI, changes in flows of one or two orders in log scale would not affect the conclusions drawn (since NFDI is a function of log of mean flows and other streamflow characteristics).

changes in biodiversity goes beyond the rationale of the original regression. Correlations are used to exploit mutual information between variables (usually proxies) to identify patterns, but do not necessarily imply causality between variables or even mechanisms of interaction. There may be many reasons why runoff and fish biodiversity exhibit mutual information across large geographic areas and evolutionary scales: factors that determine large runoff and variability patterns, such as the size and heterogeneity of freshwater habitats, do indeed influence the diversification of a fish in a basin. These processes should not be confused with short-term and small-scale effects. Smaller-scale changes in fish assemblages may indeed be related to stressors such as flow alteration, but typically in the context of combined effects of other spatially dependent factors (which may be even more important), such as reservoir effects of fragmentation and loss of access to functional habitats, water quality degradation, changes in sediments and nutrients regime, geomorphological changes, etc.

Most of the scientific literature that the authors cite to support their "intended use of the SDR that dams alter flow regimes resulting in long-term loss of diversity" actually disproves their point: these studies look spatially at multiple factors affecting fish biodiversity, and not surprisingly, none make claims as extreme and sensational as those implied by the FSR equation used in this study, of reductions >70% in fish biodiversity (~100 species) at the basin scale. Likewise, according to your recommended references, Barbarossa et al. document a positive correlation between human transformation and fish biodiversity. By the same logic, should your study argue that the current fish biodiversity in the Cauvery has been positively influenced by the reservoirs in the basin? Why cherry pick factors?

Hence, a reservoir configuration leading to substantial alterations in streamflow characteristics — deviating not just marginally but significantly from mean flows — would profoundly impact the NFDI. It must demonstrate a substantial increase in economic value to remain a Pareto superior choice. Reservoirs that significantly alter flow regimes but do not add significant value should therefore be discouraged since it would be a Pareto inferior choice.

Further, although it is acknowledged that the current analysis does not directly provide a practical solution, it highlights an important consideration for reservoir planning and management. The paper presents a proof of concept of the trade-off between the economic benefits of existing reservoirs for agricultural production and the potential negative impacts on fish diversity. While using the normalized fish diversity index as an indicator, the study provides an assessment of change in some aspects of freshwater habitat integrity. We have applied the equation developed by Iwasaki et al. (2012) and Yoshikawa et al. (2014) to the Upper Cauvery basin and have extended the application of space and time substitution based on the equation (by time here we mean the occurrence of different scenarios). The central idea is to assess how environmental quality varies with different reservoir configurations and how it trades off with agricultural production. We acknowledge the limitation of equation 5 that in explaining the variability in normalized fish diversity index it does not consider other chemical and biological factors since it is solely based on the assessment of changes in water quantity and not quality, nor of impacts of non-dam related interventions. The same holds for our model. If the impact of unaccounted variability, e.g. of water quality and non-dam related interventions, on fish species richness (FSR) exceeds the recognized reservoir-induced streamflow variability, the reliability of changes in FSR values based on Equation 5 may be

I hope at this point you realize that extrapolating the purpose of a correlation analysis is not a straightforward process as you claim. Your suggestion to rescale the FSR analysis into an "SDR-based habitat integrity" is then misleading; it aims to mask this fundamentally flawed equivalence.

Let me insist. As it stands, the study provides an assessment of changes in some aspects of freshwater habitat integrity, not fish biodiversity. I'd respectfully encourage the authors to withdraw the current submission and reformulate the paper to explore the trade-offs in those terms and avoid stretching the results to claim that this study incorporates fish biodiversity.

[revised manuscript text omitted]

| Comments | Response |
|---|---|
| This paper develops a complex methodology that combines ecological and agricultural methods to estimate the economic and environmental viability of constructing multiple dams along the Cauvery Basin in India. The paper has merit and offers a novel approach to compare economy-environment tradeoffs through the PPF. The main strength of the model lies in its hydrologic component and the capacity of the model to represent the water system, including the location of built and hypothetical reservoirs. I note throughout the methods an imbalance between the detail in the ecological analysis and that of the socio-economic analysis. The structure is adequate, and the flow of the paper is easy to follow. Not being an ecological researcher, I will refrain from assessing the merits of the ecological modeling and will focus rather on the socioeconomic component of the model.

The use of a PPF is clever and provides a useful visualization of the tradeoffs. However, the economic analysis is rather an agronomic analysis of production (ET function) to obtain yield quantities, which are later multiplied by a constant price. This approach has a number of limitations | We thank the referee for the constructive comments and look forward to any further discussion. We agree with the simplistic visualization of the tradeoff, it being the main contribution of the paper, and we have provided the clarifications as needed in our revised manuscript. |
| -Authors seem to assume that agricultural land allocation remains constant. That is, if farmers receive 50% of the water right, they irrigate each crop applying 50% of their ET and maintaining land use shares. This precludes extensive (shifting to rainfed agriculture), super extensive (shifting to another crop), and intensive margin adaptation (selective deficit irrigation, where some crops receive less water, and some others are fully irrigated). This approach lacks realism as demonstrated in the agecon literature (Graveline and Mérel, 2014). I reckon this is a very common assumption in hydrologic and eco-hydrologic models-but it is nonetheless a gross simplification that, although common in the literature, has to be properly acknowledged in the text as a key assumption and limitation of the research. Authors are invited to read about economically calibrated models and even | We agree that it is a simple model where land allocation remains constant, and we are only discussing what production is possible within the basin given this assumption. We are therefore still not engaging with utility/profit maximization since we only define our production set and calibrating utility/profit functions based on what is being currently observed is a next step. We will discuss this assumption in the light of literature mentioned.

The following line is added to the manuscript to acknowledge the limitations:

"While calculating the economic value of crops, the size of the cropped area is kept constant to isolate and analyze the impact of various |

| | |
|---|---|
| normative models used to assess land and water reallocations based on utility/profit maximization (Adamson et al., 2017; Graveline, 2016; Sapino et al., 2020). | reservoir combinations on economic outcomes. This approach simplifies the modeling process and helps in understanding the relationships and interactions between varying reservoir combinations, and crop production, without the added complexity introduced by varying land sizes. This simplification, however, comes with limitations. For example, in the face of varying water allocations, farmers can adopt various strategies related to changing crops, for example changing to rain-fed agriculture or shifting towards less water-intensive irrigated crops (Graveline & Mérel, 2014). ” **The above paragraph is incorporated into the revised manuscript between lines 774 and 779.** |
| -Authors use constant prices. This has to be properly justified. If the basin is large enough, droughts and reduced water allocations will reduce yields, which will constrain supply and can therefore significantly affect prices. Note that agricultural demand is highly inelastic so significant changes in supply may lead to abrupt changes in prices. This can partly mitigate agricultural losses due to reduced water availability during droughts (Haqiqi et al., 2023, 2022; Parrado et al., 2019). Again, this is an even more common assumption in water system models and even economic models, so this is acceptable—but should nonetheless be acknowledged as a limitation, particularly considering the economic relevance of the basin, which occupies a large territory. | The following line is added to the manuscript to acknowledge the limitations. “Another limitation of this study is the utilization of constant prices, a factor that may pose challenges in assessing the impact of droughts and reduced water allocations on crop yields. If the basin is large enough and dominates the domestic market in terms of production of certain crops, then droughts and reduced water allocations will reduce crop yields, which will constrain supply and can therefore significantly affect prices. Since agricultural demand is highly inelastic, significant changes in supply may lead to abrupt changes in prices (Haqiqi et al., 2023, 2022; Parrado et al., 2019). As agricultural markets are well-developed in the basin and well-connected to other domestic and international markets outside the basin, production changes in the basin, could be compensated by production in  neighboring places unless there is a significant supply shock”. **The above paragraph is incorporated into the revised manuscript between lines 781 and 787.** |

| | |
|---|---|
| Last, although I'm not an expert ecologist you are assessing environmental–economy tradeoffs as a relationship between fish biodiversity and agricultural production. I understand agriculture is the main and least valuable economic use, so this makes sense. But are not their other relevant ecosystem services affected by the dams? Such as aesthetic amenities, tourism, etc.? Why are you not including them? A justification here would be necessary. | We agree and have acknowledged in the manuscript that dams affect various ecosystem services and that we used agriculture because it dominates the economic value produced in the basin. We have also highlighted that a comprehensive economic and non-economic valuation of all these services is needed but requires extensive data and resources. It is also beyond the scope of the current study and is left for future studies.

**Kindly see lines 698 -703 in the revised manuscript as indicated below**

"However, intangible services were not analysed in this study. For example, humans directly consume or use both agriculture and fisheries products for food, nourishment, and employment, and to support their way of living. Both agroecosystems and fisheries provide regulating and supporting services that are crucial for ecosystem functioning and resilience. However, the human-driven ecosystem dis-service from agricultural activities can reduce ecosystem resilience and decrease service generation that are necessary for human survival. Therefore, the non-tangible ES and dis-services should also be taken into consideration using appropriate economic valuation tools in a tradeoff analysis" |
| Overall, the paper has value and potential, but clarifications are needed to put this in the context of existing socioeconomic literature. | Thank you for recognizing the paper's value and potential. We acknowledge the need for clarifications in the context of existing socioeconomic literature. And we have addressed this by providing additional context and discussing how our work contributes to the existing body of knowledge. |


Reviewer #3

The authors have improved the manuscript from the earlier version, particularly on scope, aims, description of models, and parameterizations. Thank you for providing responses to earlier comments. However, there are still critical aspects of methods, analysis and interpretation of results which should be significantly revised.

**Response**

We thank the referee for the constructive comments. Below are our responses and will provide the clarifications as discussed in our revised manuscript. We look forward to further discussing our manuscript.

**Major comments:**

| Comments | Response |
|---|---|
| 1). The authors have directly implemented empirical equation for fish species richness (FSR) as an indicator for ecosystem health as proposed by Iwasaki et al. (2012), authors further argue that in lines 536-537, that Yoshikawa et al. (2014) validated the empirical equation in 84 basins. However, that is not true! In the second study, the comparison of linear and non-linear form of FSR was done and applied for different basins. Both studies discuss the limitations and mentions that FSR varies across upstream to downstream. Yet the response on previous comment, "does a single FSR value enough for whole river network?", was not adequately answered. | We have discussed the limitations such as that the equation used is a statistical one and does not consider other chemical and biological factors since it is solely based on assessment of changes in water quantity and not quality and not of impacts of other non-dam related interventions and how it builds further on the work of Iwasaki et al. (2012) and Yoshikawa et al. (2014) in the revised version of the manuscript. |
|  | **Kindly see lines 727 – 737 in the revised manuscript as indicated below** |
| The key limitation should be well taken into consideration, that the empirical equation is not based on the actual physical processes and is derived through observed data. Hence, the regression coefficients of the equations are subject to vary based on new data and across river basins. It is also be noted that sample data on those studies are not from this river basin or the latitude of this study area. | "While using the normalized fish diversity index as an indicator, the study provides an assessment of change in some aspects of freshwater habitat integrity. We have applied the equation developed by Iwasaki et al. (2012) and Yoshikawa et al. (2014) to the Upper Cauvery basin and have extended the application of space and time substitution based on the equation (by time here we mean the occurrence of different scenarios). The central idea is to assess how environmental quality varies with different reservoir configurations and how it trades off with agricultural production. We acknowledge the limitation of equation 5 that in explaining the variability in normalized fish diversity index it does not consider other chemical and biological factors since it is solely based on the assessment of changes in water quantity and not quality, nor of impacts of non-dam related interventions. The same holds for our model. If the impact of unaccounted variability, e.g. of water quality and non-dam related interventions, on fish species richness (FSR) exceeds the recognized reservoir-induced streamflow variability, the reliability of changes in FSR values based on Equation |

Adopting the same equation is not a novel contribution. At the least, the authors should build upon the studies by Iwasaki et al. (2012) and Yoshikawa et al. (2014) and discuss implications in this basin. Additionally, authors may consider sensitivity analysis and observe how FSR varies spatially along the river or temporally over the years. But this is just a suggestion. However, authors should address this from methods to results and discussion.

5 may be compromised. Unconsidered unknown variables like human footprint and fragmentation can introduce bias (Schipper and Barbarossa, 2022)''

We have also provided a sensitivity of the conclusions derived based on the tradeoff/PPF analysis when data for different time periods are considered in our revised manuscript.

**Kindly see figure 12 in the revised manuscript as indicated below**

[Figure]

Figure 12. Illustration of production set and production possibility frontier (PPF). The PPF is the outer edge of the set, between the value of agricultural production and normalized fish diversity index. The error bars represent the variability associated with agricultural production and NFDI for different years.

We have acknowledged that equation 5 is based on observed data from a global data set (including our latitude but none in India) and the equation reports only significant coefficients that indeed may change if more data is added to the regression analysis (**see lines 566 -571 in the revised manuscript**)

Further given our motivation to use it only as an indicator of environmental quality to compare across the scenarios, we have rescaled it and allude to it as an "Normalized species diversity Index,"

| | |
|---|---|
| | so that it is used only as a means to compare one scenario with the other in relative terms of the provision of an ecosystem service rather than FSR ( **Kindly see lines 409-421 in the revised manuscript** ) |
| 2). Line 18, 28, and overall: "Spatial configuration" term is still vague and might be confusing to readers. Spatial configuration means finite number of configurations of existing four or more dams (if added) to other feasible locations, it could also mean changing the storage capacities of the dams (or changing the locations of 4 existing dams). The authors did not do this but claimed this can be done (as mentioned in lines 82 and 319). The 16 scenarios are indeed created by strategically deleting one, or combination of two, three or four dams. "Configuration of existing dams" is more clear phrase. However, it is appropriate to write that the strength of this approach is it can be extended to full spatial configuration of dams (i.e., addition/deletion of new/old dams at existing/new locations etc.). | We have changed the terminology to "configuration of existing dams."

In total 21 replacements have been done.

However, could be noted that we have already highlighted the strength as suggested that it can extended to the full spatial configuration of dams. |
| 3). Previous comment on "calibration metric was used as mm/day for streamflow." to which the authors response was, "The Flex-Topo model operates on a daily time scale, and it relies on forcing data expressed in mm/day. Consequently, when calibrating the model's simulated discharge, the observed discharge in cubic meters per second (cumec) is converted into mm/day".

It is confusing to me, since [L/T] is the unit of velocity. Did the authors consider cross-sections of the river to convert discharge to velocity? Otherwise, how mm/day for discharge was obtained? Please explain this. | The mm day$^{-1}$ value of the observed and modelled discharge is obtained by dividing the volumetric flow rate by the area contributing flow to the gauge station.

**We have added this clarification in lines 236 -239 of the revised manuscript.**

"The mm day-1 value of the observed and modelled discharge is obtained by dividing the volumetric flow rate by the area contributing flow to the gauge station. The details of the parameters calibrated for the FLEX-Topo model and the reservoir operation model are provided in Supplementary materials. Also, the NSGA-II parameter setting are detailed in the Supplementary materials; see Tables S.1, S.2 and S.3" |

| | |
|---|---|
| 4). Previous comment on, "Figure 6: It is redundant with Table 2 (Mean annual flow column). Presenting hydrographs would be more informative than Figure 6 since there are only 16 scenarios and expanding the discussion on the role of reservoirs and seasonal streamflow." to which authors responses was, "The Table 2 is deleted, and the figure is modified to hydrograph as suggested."
 Figure 6 is not the hydrograph! It is in fact same figure as previous version of Figure 6. Authors just changed horizontal bars to vertical.
 My suggestion is mean annual flow column should not be removed from table 2. And hydrographs of mean monthly flow (Jan to Dec), or similar figure should be added to discuss seasonality, change in minimum flow, peak flow etc. In addition, authors presented the time periods for calibration and validation. | I believe that this is a mistake made in our response to call Figure 6 a hydrograph. We have removed mean annual flow column from Table 2 and replace current Figure 6 with mean monthly flows for all 16 combinations. `

 Kindly see Figure 6 in the revised manuscript

[Figure]

 Figure 6 . The mean monthly flows resulting from different configurations of reservoirs from 2011 to 2016

 **The figure is indicated in line 455  of the revised manuscript** |
| But what is the time period the analysis of 16 scenarios? | The time period for the analysis of 16 scenarios is 2011-2016 due to availability of relevant agricultural data. |
| 5). I am confused with how the authors compared high value crops for basin (of smaller dams), with less value crops for basin (with larger dams). It is difficult to understand why | We agree that cereals may have more intrinsic value to farmers, even if it has low market value, than the monetary value offered by horticulture crops. What we have assumed is that farmers can always buy cereals in the market at prices lower than the prices at which they sell horticulture crops |

| | |
|---|---|
| such comparisons are made. May be the cereal has more value compared to horticulture crops for livelihood of the community, although the monetary value may be opposite. I am not sure how appropriate it is to recommend switching to different crops just for monetary value, since water management is multi-stakeholder decision, not just driven by monetary gain. | and still save. Therefore, implicit is the assumption that markets are well connected to other domestic (out of basin) and international markets and local scarcity of certain crops does not significantly affect its prices.

 **We have clarified this in the lines 785-787 of the revised manuscript and it is indicated below**

 "As agricultural markets are well-developed in the basin and well-connected to other domestic and international markets outside the basin, production changes in the basin, could be compensated by production in neighboring places unless there is a significant supply shock" |
| 6). Section 2.7.1: Since, the model was simulated at daily time steps, how was the agricultural production computed? Equation 4 do not describe this. Perhaps you can add another equation to elaborate how it was implemented in the model, and how the yearly agricultural production (as shown in figure 10) was computed. Please indicate the correct units of the variables implemented in your model. | We have added the explanation as suggested, also equation 4 is missing a summation size in front of the ET ratio which we have modified.

 **Kindly see lines 352-355 in the revised manuscript**

 "The equation 4 presents end-of-season yield as a fraction of optimal yield that depends on how much daily evaporation is accumulated by the crops over the season compared to the respective evaporation demands (optimal evaporation). Yearly production value is obtained by multiplying the average area of each crop with average simulated yields and prices over 2011-2016" |
| 7). Equation 5: Similar to the comment on equation 4. But is this computed once per year? Does it vary each year? How the single value of FSR as shown in Figure 11 was obtained over the time period of simulation, assuming it was a multiyear simulation? | **The following text is added and updated in lines 424-427 of the revised manuscript**

 "Daily-scale simulations for calculating FSR parameters like TH3, FL2, and TL2, along with mean annual flow calculations and evaporation deficit in yield estimations. Daily-scale modeling facilitates space-time substitution in SDR-based FSR, enabling assessment of agricultural production trade-offs with reservoir combinations. In these scenarios, other factors are assumed constant". |
| 8). Line 15: It is a unique approach or unique for the Upper Cauvery basin, India, or first study using this approach in this basin. Agriculture value and fish species richness was quantified as a single value per scenario not analyzed at daily scale, so the statement is incorrect. Also, how the use of daily scale is unique contribution? | To our knowledge, such a tradeoff analysis is unique to India based on model simulations that incorporate model simulations at a daily scale. Daily scale simulations are necessary for the analysis presented because both FSR and agricultural production need calculations at a daily scale.

 **The following text is added and updated in lines 424-427 to the revised manuscript** |

| | "Daily-scale simulations for calculating FSR parameters like TH3, FL2, and TL2, along with mean annual flow calculations and evaporation deficit in yield estimations. Daily-scale modeling facilitates space-time substitution in SDR-based FSR, enabling assessment of agricultural production trade-offs with reservoir combinations. In these scenarios, other factors are assumed constant". |
|---|---|
| 9). Line 60-70: Upon quickly verifying these by google search, this list is not updated. There are several more studies on cascade or multiple dams – and its environmental implications. | We agree, our intention was not to cite all recent studies on multiple dams. Few more relevant studies on multiple dams have been added to the literature as indicated below :

 "Several studies have targeted multiple dams (Consoli et al., 2022; Van Cappellen and Maavara, 2016; Ouyang et al., 2011; Wang et al., 2019; Zhang et al., 2020; He et al., 2022)"

 **Kindly see lines 62-64 in the revised manuscript** |
| 10). Line 67-69: What are pre-dams data? Please check the validity of this statement. | The sentence has been modified as indicated below

 "However, there are no studies that assess the impact of multiple dams on the provision of ecosystem services at macro basin scales and at daily time step when pre-dams data is unavailable"

 **Kindly see lines 67-68 in the revised manuscript**. |
| 11). Figure 1. Calibration can be omitted from this flowchart since it was done in a previous paper and not part of this paper. | Modified as suggested, **see figure 1** |

[Figure]

| | |
|---|---|
| **Other comments:** | |
| 12). Section 3.1: Since, calibration and validation are not part of this study, but was taken from previous study, it is not necessary to present it in result section. A separate section on "Description of model" or merging it in introduction with supporting supplementary material would be better. | We will refer to the supplementary material for calibration and validation and we have removed section 3.1 (since similar text also appears in the supplementary material) |

| | |
|---|---|
| 13). Equations 1, 2, 3: Please write the units of the variables used in descriptions. | The units of the variables were added which were used in the description. **Kindly refer line 206-208, 234-236 and 347-349 in the revised manuscript** |
| 14). Figure 3, legend on size of point symbols can show the range of minimum to maximum area of command area. Another question, is size of the command area reflective of the number of populations living in that area, or the size of irrigated land? If not, what is the relevance of the size of command area! | Kindly note that a command area of a reservoir refers to the irrigation area commanded by the reservoir. We have defined the command area in page 6. **See foot note in the caption of Fig.2 (line 131):** "[1] A command area is the area which can be physically irrigated from a reservoir and is fit for cultivation" |

---

## Author Response (AR2)

**Response to the referees**

We thank the referee for the constructive comments. Please find our point by point response to the comments below:

1. I think the major issue is with the claim that study propose a novel method e.g., in abstract (line 26) and in (line 791) conclusion. However, the presentation of the methods and analysis suggest retreating from that assertion. Specifically, the claim of novelty outlined in line 80-82 - using a synthetic configuration of reservoirs to study hydro/ecological impact - does not constitute a novel approach. The authors should reassess their claims regarding the novelty of their method and accurately articulate the realistic contributions of their work.

**Response:** Based on our literature review there is a gap in assessing the impact of multiple dams on the provision of ecosystem services *at daily time step when pre-dams data is unavailable* (lines 68-69) and this paper fills this gap – hence by doing so we highlight that the approach is novel (in addition to plug and play approach that allows on top of this a means to assess synthetically based dams anywhere).

We have now emphasized that "The approach is novel not only because it offers the analysis of alterations in ecosystem services at daily scale when pre-dam data in unavailable but also because dams can be synthetically placed anywhere in the river network and the corresponding alterations in flow regimes simulated in a flexible manner." (Lines 13-16) and made adaptations also in Line 791. In the same go we also clarify that while the approach can deal with synthetically placed reservoirs, this paper only works with combinations of existing reservoirs as a proof of concept and a first step in demonstrating the approach (e.g. see line 16). Thus the changes are now aligned with the methodology implemented in the paper.

To avoid repetition and any misunderstanding, we have removed the word novel from line 80-82.

2. Some of the statements made are vague, not specific, and lack convincing backing from data or information. Example, (Line 67-68) "However, there are no studies that assess the impact of multiple dams on the provision of ecosystem services at macro basin scales and at daily time step when pre-dams data is unavailable"

**Response:** Our literature review suggests there are no such studies at daily scale when no pre-dam data is available. We have therefore adapted the statement as:

"However, our literature review suggests that there are no studies that assess the impact of multiple dams on the provision of ecosystem services at daily time step when pre-dams data is unavailable."

3. Lines 13-15 sounds like a novel approach was proposed, but as per the response it is unique application for this particular basin only, which should be corrected.

**Response:** please see our response to comment 1.

4. The use of normalized index is more appropriate choice than using FSR. In Figure 12 Please explain what the error bars represents (is that maximum - minimum or quantiles?). Again, what does the intersection points represent?

**Response:** The error bars around the mean values of agricultural production and NFDI, for a given reservoir scenario, are one standard deviations of the agricultural production and NFDI simulations for the reservoir scenario over different years. We have adapted the caption of the figure accordingly, please see lines 603-605.

5. Lines 236 -239: It is still unclear what is the motivation for selecting [L/T] as unit of discharge for calibration while other figures are shown in discharge unit. In table S.3, MAE for reservoir is shown in cubic meter per day but MAE for sub-basin is shown in mm per day. Why divide the volumetric flow by area?

**Response:** This was an error, the MAE units for reservoir are also in mm/day and not cub m per day. We have now corrected the units.